# Landslide topology uncovers failure movements

Kushanav Bhuyan [1,2,8] ✉, Kamal Rana [2,3,4,8] ✉, Joaquin V. Ferrer [4,5], Fabrice Cotton[2,6], Ugur Ozturk [2,4], Filippo Catani[1] & Nishant Malik [7]

The death toll and monetary damages from landslides continue to rise despite advancements in predictive modeling. These models' performances are limited as landslide databases used in developing them often miss crucial information, e.g., underlying movement types. This study introduces a method of discerning landslide movements, such as slides, flows, and falls, by analyzing landslides' 3D shapes. By examining landslide topological properties, we discover distinct patterns in their morphology, indicating different movements including complex ones with multiple coupled movements. We achieve 80-94% accuracy by applying topological properties in identifying landslide movements across diverse geographical and climatic regions, including Italy, the US Pacific Northwest, Denmark, Turkey, and Wenchuan in China. Furthermore, we demonstrate a real-world application on undocumented datasets from Wenchuan. Our work introduces a paradigm for studying landslide shapes to understand their underlying movements through the lens of landslide topology, which could aid landslide predictive models and risk evaluations.

Landslides cause economic damages worth 20 billion US dollars every year[1], and between 2004 and 2019 non-seismic landslides alone caused about 70,000 fatalities worldwide[2]. Within the first two months of 2024, we have seen reports of devastating landslides in Colombia[3], Southern Philippines[4], and Yunnan, China[5], injuring many and killing approximately 74 people. Adding to this, recent studies count over one million landslide occurrences, with annual volumes estimated at fifty-six billion cubics meters globally[6], presenting a risk to sixty-five million people[7]. With the increase in urbanization, global climate change, and environmental change trends, the frequency of landslides and the associated risks will keep increasing globally over time[7]. In line with this, landslides are anticipated to evolve and remobilize with increased frequency under changing climatic conditions on a decadal scale[8,9]. Our ability to identify hazards from emerging landslides and

dynamically assess impact areas is essential in averting risk to rapidly urbanizing communities and adapting to changing environmental conditions[7,10].

To address the rising landslide risk, predictive models for hazard, risk, and early warning systems assist in forecasting landslide occurrences and locating landslide-prone regions to mitigate the undesired impacts[11]. However, the efficacy of these models is contingent on the quality of the underlying landslide databases. These databases often lack the much-needed information about the type of failure of the mapped landslides[12]. Generally, these databases include a broader definition of landslides that covers all types of gravitational mass wasting processes, such as slides, flows, and falls including sub-types based on their movement, e.g., rotational slides[13] combined together, thereby hampering the capability of predictive models. Typically, each

[1]Machine Intelligence and Slope Stability Laboratory, Department of Geosciences, University of Padova, Padova 35129 Veneto, Italy. [2]Helmholtz Centre Potsdam - GFZ German Research Centre for Geosciences, Potsdam 14473 Brandenburg, Germany. [3]Chester F. Carlson Center for Imaging Science, Rochester Institute of Technology, Rochester 14623 NY, USA. [4]Institute of Environmental Science and Geography, University of Potsdam, Potsdam 14473 Brandenburg, Germany. [5]Potsdam Institute for Climate Impact Research, Potsdam 14473 Brandenburg, Germany. [6]Institute of Geosciences, University of Potsdam, Potsdam 14473 Brandenburg, Germany. [7]School of Mathematics and Statistics, Rochester Institute of Technology, Rochester 14623 NY, USA. [8]These authors contributed equally: Kushanav Bhuyan, Kamal Rana. ✉e-mail: kushanav.bhuyan@phd.unipd.it; kr7843@rit.edu

landslide failure type exhibits different geological, geometrical, and geotechnical properties (see Fig. 1). For instance, slides have conspicuous primary scarps and collapse along the planar or rotational surfaces[14], flows such as mudflows exhibit visco-plastic or viscous/fluid kinematics caused by excess pore water pressure[15], and rock falls entail the free falling of fragmented rocks from steep slopes[16] (see Note S1 and S2 for detailed explanations). Practitioners usually combine these different failure movements into one group within an inventory, despite their different properties[17–19], since categorizing them manually requires comprehensive surveys (both in the field and remotely) and standardized classification protocols[12], which are laborious and time-consuming. Consequently, predictive models start to harbor significant levels of uncertainty and bias[20], hence failing to match empirical observations, especially when moving from local levels to regional and global scales[21–23].

Preliminary attempts at identifying failure movement types have considered both knowledge-driven and data-driven approaches. While the former are region-specific, bounded by expert-based rules, and constrained to small areas[24,25]. The latter addressed these problems with supervised learning and have successfully identified landslide failure types in the Italian context[26]. However, the existing solutions are limited in their prediction capabilities, as the failure information is derived from geometric properties of two-dimensional (2D) landslide polygons (outlining the landslide planforms). Owing to the inherent limitations of 2D landslide polygons, crucial kinematic and mechanical details embedded in the landslides' three-dimensional (3D) morphology are overlooked, such as the style of kinematic progression, deformation patterns and structures, and depositions. The complex kinematic evolution of one or more failure types may culminate in the convergent evolution of landslide shapes, wherein landslides starting as completely different movements may evolve to follow similar planar outlines (debris slides converging to debris avalanches[13]). This presents challenges in deducing failure types solely based on 2D

geometric descriptors or rudimentary topographic metrics. We argue that these overlooked kinematic attributes are intricately linked with the 3D morphological properties of landslides, which can be comprehensively analyzed through topological methods.

Topology is a sub-discipline of mathematics explored in many fields that concern the study of shapes[27], such as in protein structures[28], data modeling[29], complex networks[30], and signal processing[31]. We explore advanced data analysis tools rooted in topology, known as topological data analysis (TDA), which captures critical structures present in the data's shape (in our case, landslide's 3D shape). We hypothesize that key features of landslide kinematics are embedded in the 3D topology of the involved landforms and that TDA properties can capture their kinematic movements as a proxy for identifying landslide failures.

In this study, we introduce an approach for uncovering landslide failure types based on their mode of movement by examining the topological properties inherent in the 3D shapes of landslides. We develop the method using the Italian historical inventory from Inventario dei Fenomeni Franosi (IFFI)[32] and then deploy it to landslide inventories from varying geomorphological and climatic settings: the United States (US) Pacific Northwest region (which includes the states of Oregon and Washington), Denmark, Turkey, and Wenchuan, China (see Fig. 2) to validate the effectiveness and applicability of the approach. We demonstrate that the method offers a more comprehensive understanding of the underlying failure types—slides, flows, falls, and complex—compared to traditional analyses based on 2D polygonal geometry, as hypothesized. We explore the model's ability to identify sub-types of failure movements (e.g., translational and rotational slides, earth and debris flow). Also, we utilize the topological properties of complex landslides to reveal coupled failure types underlying the formation of complex landslides. In addition, we identify the types of failure in an event-based multi-temporal landslide inventory, enhancing our understanding of landslide dynamics and

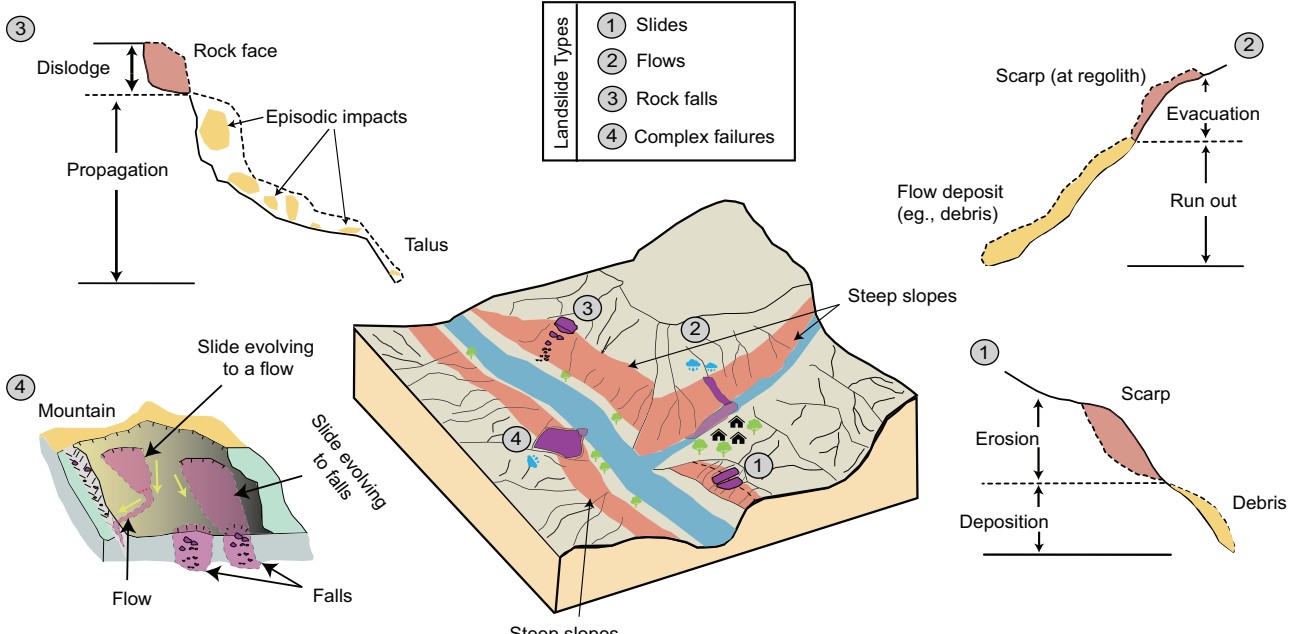

**Fig. 1 | A schematic representation of different landslide failure movements.** The schematic broadly illustrates different landslide failure types and their associated mechanical and kinematic behavior, excluding sub-type movements. For example, Type 1 refers to slide-type failures that can constitute deep ruptures, where a cohesive unit of soil or rock (colored in red) slides down a slope following a well-defined rupture plane (colored in yellow). Type 2 refers to flow-type failures where regolith, rock, or other material travels down a slope as a dense, fluid-like mass with a flow-like motion. Type 3 is a fall-type failure where a body of rock detaches from a steep slope or cliffs and exhibits free-falling and episodic impacts as it propagates down the slope. Type 4 refers to the complex interaction and effect of numerous geomorphic processes transpiring in a single failure event, where processes start as one type and evolve into another; such as a slide-type failure evolving to a flow-type failure.

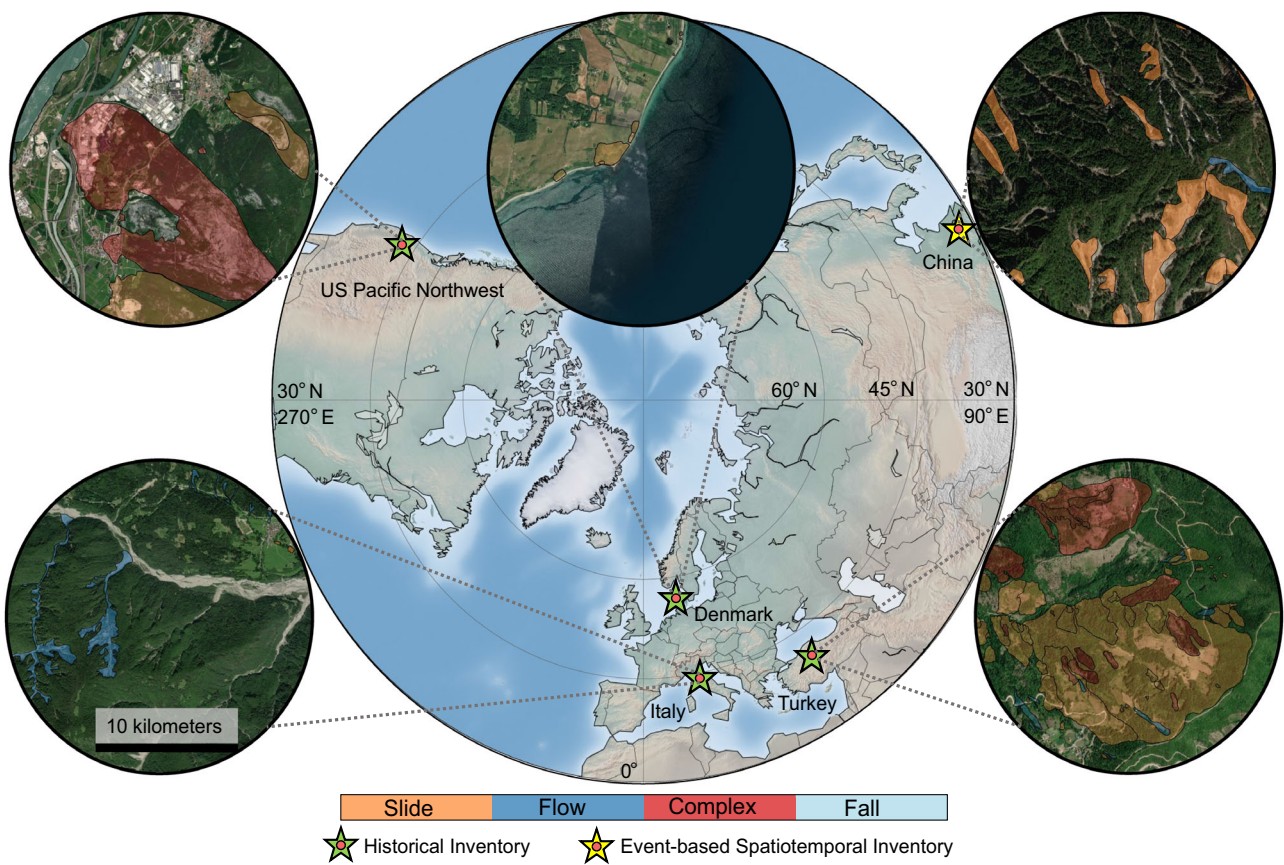

**Fig. 2 | Location of the study areas.** The diagram shows the geographical regions: Italy, the US Pacific Northwest, Turkey, Denmark, and Wenchuan, China whose data we analyzed in this work. The green star showcases regions encompassing historical inventories and the yellow star showcases an event-based spatiotemporal inventory. The inset in the circular shape shows snippets from diverse regions with landslide polygons of different failure movements (slide: colored in orange, flow: colored in dark blue, complex: colored in red, and falls: colored in light blue) on top of the World Imagery from ESRI. Map credits: Esri, Maxar, Earthstar Geographics, and the GIS User Community[68].

behaviors in real-world scenarios. Moreover, we deployed our method on two undocumented (paleo-event) datasets as a real-world application, which we verified using Google Earth archive imageries. To the best of our knowledge, sub-type failure movements, temporal prediction of failure movements, and detection of the underlying failure types within complex landslides have never been investigated using an automated data-driven approach. Here, we showcase with our findings that the proposed method (1) is user-friendly, exclusively requiring only the landslide polygonal shape and a Digital Elevation Model (DEM) as input, (2) exhibits high performance in discerning failure types based on their movements, (3) is adaptable across various geomorphological and climatic regions, and (4) shows strong performance and remains robust, even with limited training samples, thus indicating the method's deployability in data-scarce regions. By offering a deeper understanding of landslide failure movements, our approach has the potential to enhance the accuracy and reliability of landslide susceptibility, hazard, and risk assessment models, providing valuable insights to the predictive modeling community.

## Results

### Landslide topology as a proxy to identify failure types based on style of movement

The underpinning of topological data analysis (TDA) is rooted in structures in the data's shape, such as connected components and holes. Holes represent the empty spaces in the data's shape, and connected components represent the connection of the data's points linked by a continuous path[33]. Using the holes and connected components, we can calculate various topological properties to quantify a shape. For this, we perform the TDA on landslide shapes to compute topological properties, which can then be used as a proxy to investigate the underlying failure types. It is important to note that we employ 3D point clouds (containing geographical latitude, longitude, and elevation information) from the landslide's outline (see Fig. 3) estimated via the landslide polygon and the digital elevation model (DEM). The landslide polygon provides the best available approximation of the landslide boundaries in the geographic space, as derived by standard surveying methods with suitable accuracy.

The degree of compactness in a landslide shape is essential when identifying failure types[26]. For instance, slides are characterized by a more cohesive material that tends to remain as a single component (e.g., slides with clay-rich soil[34]), leaving behind a more compact-shaped footprint as they fail. In contrast, flow-type failures involve more fluid and fragmented materials deposited on a debris fan and display viscous/fluid kinematics that follow the channelized topography of the natural landscape and are hence, sinuous and less compact. On the other hand, falls consist of fragmented materials that roll or bounce off steep cliffs with a rather shorter and straighter run-out path compared to flows, which leave behind a footprint that has an intermediate degree of compactness between the slide and flow-type failures.

We use the amount of empty space inside the footprints of the landslide shape outline to quantify its compactness. To give a simple example, a higher amount of empty space in a landslide shape outline is associated with a higher degree of compactness (e.g., for slide-type failures, as seen in Fig. S1a). Representing the average lifetime of holes, $AL_H$ (one of the topological properties) computes the hole's average size and estimates the information pertaining to the

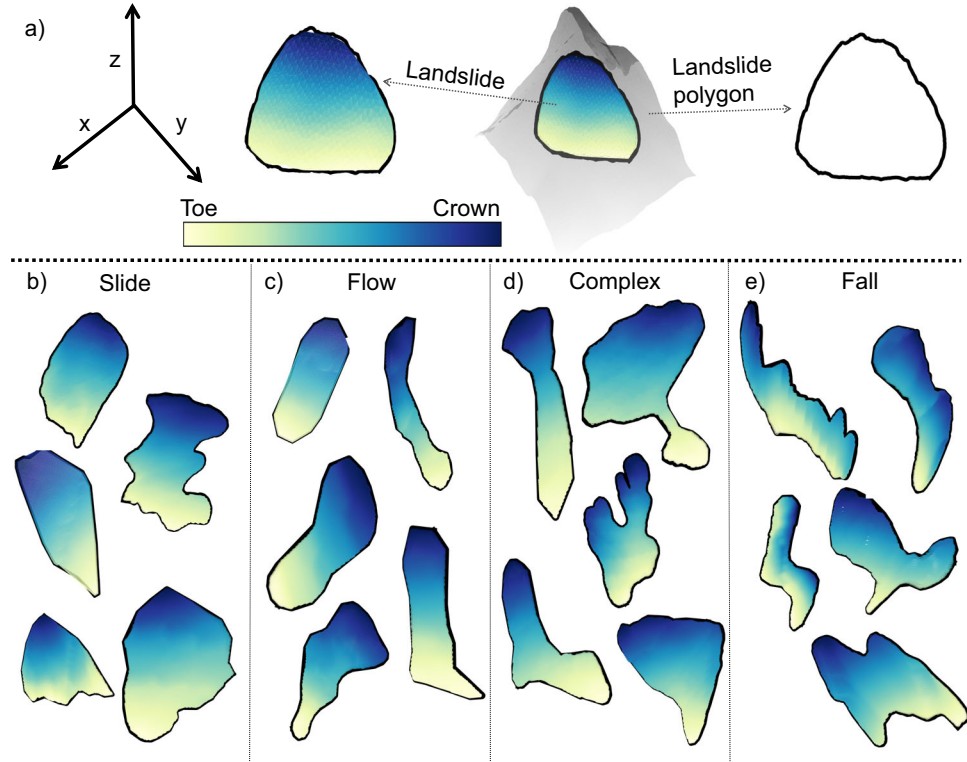

**Fig. 3 | 3D illustration of different landslide movements and their polygonal shapes. a** An example of a landslide failure in a terrain with a steep slope. The diagram also shows the 3D landslide polygon, which outlines the landslide shape (depicted with a gradient color from dark blue to light green highlighting relative elevation from the crown to the toe). **b–e** 3D landslide samples for different landslide failure types, namely, the slide, flow, complex, and fall types.

empty space, and thus the compactness of a landslide's shape. So, landslide shapes with a longer $AL_H$ are more compact than shapes with a shorter $AL_H$. Based on the probability density function (PDF) of the $AL_H$, our analysis reveals that slides are more compact than flows, falls, and complex landslides because they have a longer $AL_H$ (see Fig. 4). We observe the PDF of the $AL_H$ curve for falls to lie between those of slides and flows, showing that empty spaces generated in falls do not survive long since materials detach from steep slopes and travel a short distance, thereby leaving behind a footprint that represents an intermediate level of compactness. Also, the PDF of the $AL_H$ for complex landslides shows an intermediate level of compactness, credited to their amalgamated behavior as a combination of slides, falls, and flows.

Another critical property for diagnosing failure types is the sinuosity of the transport zone, which describes the landslide's path or kinematic propagation as it progresses downslope. Among various types of failure, flows exhibit the greatest sinuosity, following the channelized topography more than other mass-wasting mechanisms. This pattern is attributed to the fluid and mobile characteristics of the materials involved. In contrast, slides are the least sinuous as their deposits are rarely channelized and remain on the open slope, resulting in a relatively straight and uniform path. Fall-type failures are comparatively less sinuous than flows but still exhibit some degree of sinuosity, as they too follow the landscape's topography. Sinuosity defines the existence of numerous curves in the landslide's shape attributed to the landscapes' topography, leading to the generation of partitions within the landslide outlines by the TDA and, hence, generating multiple empty spaces with a shortened lifetime (see Methods section). This information on the sinuosity of landslide shapes is inferred from the combination of two topological properties—the bottleneck amplitude of holes, $BA_H$, and the average lifetime of holes, $AL_H$. The $BA_H$ represents the maximum lifetime of holes in the landslide shape, which quantifies the maximum empty space in the 3D space occupied by the landslide. As sinuous shapes result in numerous smaller empty spaces with shorter lifetimes, the $AL_H$ drops without significantly impacting the largest empty space as determined by the $BA_H$ (see Fig. 4). Consequently, a landslide shape with a relatively higher $BA_H$ and a shorter $AL_H$ is indicative of increased sinuosity. In light of these observations, our findings indicate that flow-type landslides indeed display a higher degree of sinuosity compared to other failure types. This finding is evident from the fact that flows exhibit a similar $BA_H$ to falls but a shorter $AL_H$ in comparison. This is expected, as flows, being the most sinuous, cause multiple small holes or empty spaces (see flow-type and fall-type failures in Fig. S1b, c) that lead to shortening the $AL_H$. Conversely, slides display both longer $BA_H$ and $AL_H$, reflecting their minimal level of sinuosity.

We are also interested in the role of slope variations, as they significantly impact the stability of the hillslope and influence the type of landslide. For example, falls and slides have a more significant slope transition in their profiles compared to flows, which propagate with a nearly constant slope[35]. This slope variation is captured by the lifetime of the connected components. A sharp change in the hillslope causes the points outlining the landslide to be spaced vertically further apart, leading to a longer lifetime of the connected components. Two topological properties–the Wasserstein amplitude of the connected components, $WA_C$, and the average lifetime of the connected components, $AL_C$–help capture information about this slope variation in a landslide's profile. The $WA_C$ quantifies the set of longer lifetimes of the connected components, quantifying the most significant slope change in the landslide outline. This quantification is nicely illustrated in the PDF (Fig. 4) of $WA_C$, which shows that slide and fall failures underwent more drastic slope changes compared to flows. Yet, falls possess a shorter $AL_C$ than slides. This is due to the lower portion of the shape's outline (at the talus) displaying a flatter terrain (representing the area where materials accumulate) and attributing negligible slope change, which ultimately shortens the $AL_C$. In contrast, flows display the minimum $AL_C$, as they roughly propagate on constant slopes.

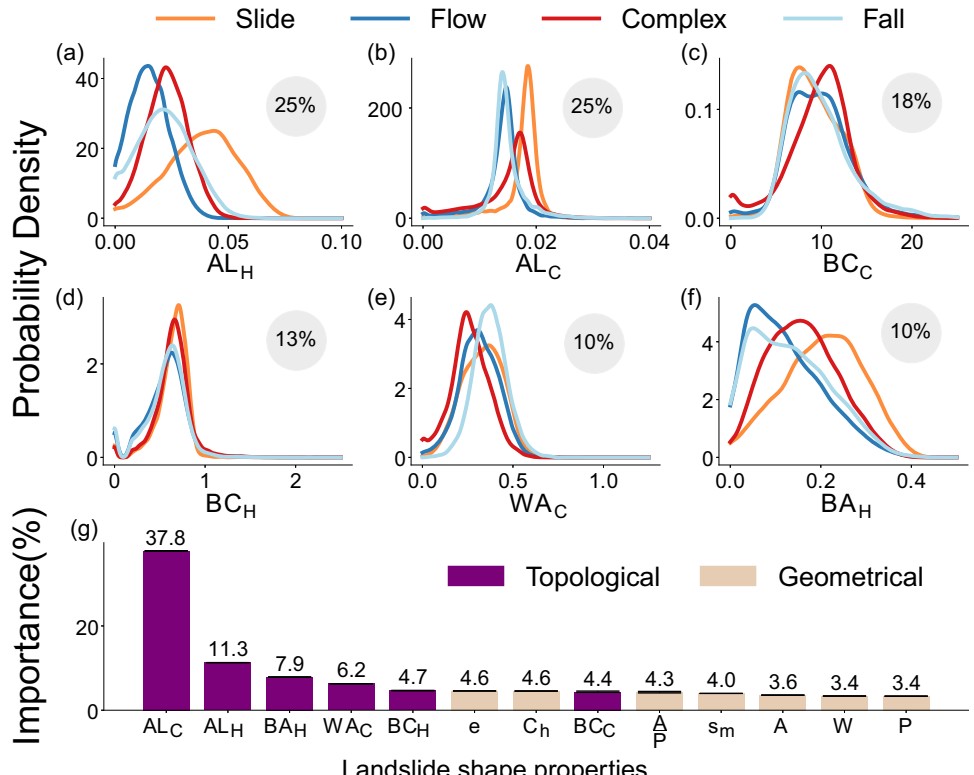

**Fig. 4 | Feature statistics and importance of landslide topology and geometry.** Plots **a**–**f** show the probability distribution functions of the six most optimal topological properties used in classifying the failure types for slides (colored in orange), flows (colored in dark blue), complex (colored in red), and falls (colored in light blue) in Italy. Note that we discuss the probability distribution functions of different failure types for the Italian region only, as the Italian data set is the most data-rich inventory. The *y*-axis shows the probability density values (calculated using kernel density estimation), and the *x*-axis shows the value of topological attributes. The topological properties in plots **a**–**f** are: Average lifetime of holes ($AL_H$), Average lifetime of connected components ($AL_C$), Betti-curve based feature of connected components ($BC_C$), Betti-curve based feature of holes ($BC_H$), Wasserstein amplitude of holes ($WA_H$), and Bottleneck amplitude of holes ($BA_H$) (the computations of these properties are explained in detail in Note S3). The percentage values in the gray circular disk in each figure indicate the topological feature's importance (in %), as estimated by the random forest-based classification procedure. Plot **g** shows the joint computed feature importance of topological (colored in purple) and geometric (colored in beige) properties by the random forest model. The analysis shows topological properties consistently outperform geometric properties with a standard deviation under 0.1% (the error bar represents the standard deviation). However, it is not visible in plot-**g** because the standard deviation is very small). The geometric properties are: area ($A$), perimeter ($P$), the ratio of area to perimeter $\frac{A}{P}$, convex hull-based measure ($C_h$), minor ($s_m$) (refer to Note S4 for the definitions), and width ($W$) of the minimum area bounding box fitted to the polygon.

Several topological properties, like the Betti curve-based feature (BC), capture more intricate landslide shape properties and help in discerning landslide failure types. The Betti curve-based feature represents the total number, lifetime, and presence of the structures (holes and connected components) emerging simultaneously. We hypothesize that it encompasses a combination of compactness, sinuosity, slope variations, and similar structures within a given landslide shape. However, the exact connection to the underlying physical types is not clear due to the complex nature of this topological property. We anticipate that such properties consider higher-order information about the landslide shape that is not immediately apparent.

Through our analysis, we discovered that common topological properties such as $AL_H$, $AL_C$, and $BC_C$ reflect the general movements of distinct failure types. These properties act as proxies for the diverse kinematic and mechanical characteristics, essential to consider when identifying the different failure types. This finding simplifies and brings coherence to our understanding of landslide behaviors through a topological lens, offering a more effective approach to discerning and predicting complex phenomena.

### Advantages of landslide topology over landslide geometry
Traditional geometric descriptors of landslide shape, including properties such as area, perimeter, and convexity, are derived from 2D representations of the landslide body[36]. As a consequence of this inherent simplification, these 2D-based geometric properties may not adequately capture crucial information, such as failure depth or internal deformations, associated with the landslides' 3D configuration. To address these limitations and provide a more comprehensive understanding of the landslide dynamics, we computed topological properties that are derived from the landslides' 3D configurations. We postulated that the topological properties would prove more meaningful in decoding the characteristics of the landslides and their underlying failure types than the traditional geometric counterparts. To test this, we used a set of seven well-known geometric properties that are commonly employed in the literature[36–38] along with six topological properties—average lifetime of holes ($AL_H$), Average lifetime of connected components ($AL_C$), Betti-curve based feature of connected components ($BC_C$), Betti-curve based feature of holes ($BC_H$), Wasserstein amplitude of holes ($WA_H$), and Bottleneck amplitude of holes ($BA_H$)—(for the justification of using six topological properties, please refer to the Methods section) to determine the failure types in Italy.

We jointly computed the feature importance of geometrical and topological properties using the Gini-index feature importance method in the random forest algorithm (see Fig. 4g). After running over 100 iterations on the Italian inventory, our findings consistently demonstrated that topological properties exhibited higher feature

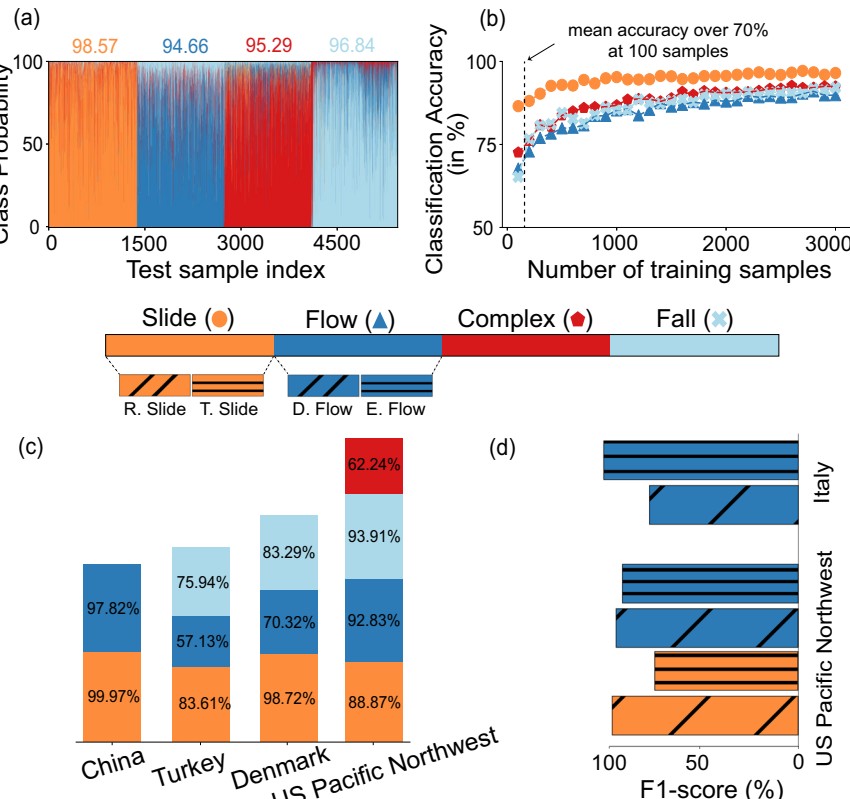

**Fig. 5 | Analysis of landslide classification accuracy across regions and sub-movement failures.** Plot **a** shows the classification accuracy (in %) for each failure class in Italy (slides: colored in orange, flows: colored in dark blue, complex: colored in red, and falls: colored in light blue). The *x*-axis of the plots shows the testing sample's index, and the *y*-axis shows the class probability corresponding to each failure class. Plot **b** shows the classification accuracy (in %) corresponding to each failure type with the number of training samples. The *x*-axis shows the number of training samples from each class used to train the model, and the *y*-axis shows the classification accuracy (in %) corresponding to each class. At 100 samples, the mean classification accuracy already reaches over 70% in Italy. Plot **c** demonstrates the model's versatility in accurately identifying various types of landslides across

multiple geographical regions, including Wenchuan, China, Turkey, Denmark, and the US Pacific Northwest, with their corresponding F1-scores listed for each type. Plot **d** highlights the model's capability to distinguish sub-types of landslide failures, specifically in the US Pacific Northwest and Italy. In the US Pacific Northwest, the model successfully classifies four additional sub-types--rotational slide (colored in orange with diagonal strips), translational slide (colored in orange with horizontal strips), debris flows (colored in dark blue with diagonal strips), and earthflows (colored in dark blue with horizontal strips)—with an average F1-score of 84% along with the other failure type classes (complex and fall type). In Italy, the model identifies two additional sub-types—debris flows and earthflows—with an average F1-score of 96% along with the other failure movements (slides, complex, and falls).

importance than their traditional geometric counterparts (achieving Micro F1-scores of 94% and ~65% respectively), yielding superior predictive capabilities for identifying failure types. Additionally, we observed that even the least important topological property ($BC_C$) has similar feature importance as the other geometric ones, while the former conveys unique information about the landslide shapes as discussed in the previous section. Moreover, we calculated the probability density function (PDF) for both geometric and topological properties and observed that the latter had greater dissimilarity than the former among different failure types (see Fig. S2). These two findings demonstrate that topological properties are stronger predictors for identifying failure types. The reason behind this can be attributed to the enhanced capacity of topological properties to encapsulate important information pertaining to landslide kinematic progression, failure depth, sinuosity, compactness, and slope variation.

### Determining failure types with TDA and machine learning

Next, we employed topological data analysis (TDA) to compute a diverse array of topological properties/features using the landslide inventory of Italy. Subsequently, we conducted a correlation analysis and feature importance assessment to identify the six most optimal properties out of thirty. Our evaluation unveiled that several TDA properties were redundant, amplifying the model's complexity and

undermining its predictive potential. Consequently, we opted to eliminate the irrelevant ones. Leveraging the six best features, we applied the random forest algorithm to discern the failure types. We independently scrutinized the performance of our approach using various training and testing sets for the Italian inventory.

We applied the model to approximately 250,000 landslide samples, ensuring balanced training by using an equal number of samples −13,000 for each landslide type. To mitigate over-fitting and bias, we performed 10-fold cross-validation 1000 times on a subset of 54,440 samples. This approach yielded a Micro F1-score, a key performance metric, surpassing 94% for each failure type (see Fig. 5a), with a performance standard deviation of less than 0.2%. This illustrated the robustness of our methodology in handling variations among training samples across Italy. We examined various other metrics, such as the true positive rate (TPR) and true negative rate (TNR), to evaluate the method's performance. These metrics consistently exhibited high scores (94–98%) across all classes, thereby ascertaining the model's classification ability.

### Method implementation for failure-type identification

In this sub-section, we assess our method's implementation across both historical and event-based landslide inventories that include both spatial and temporal data. Additionally, we tested the method's efficacy in scenarios where information on failure types is either

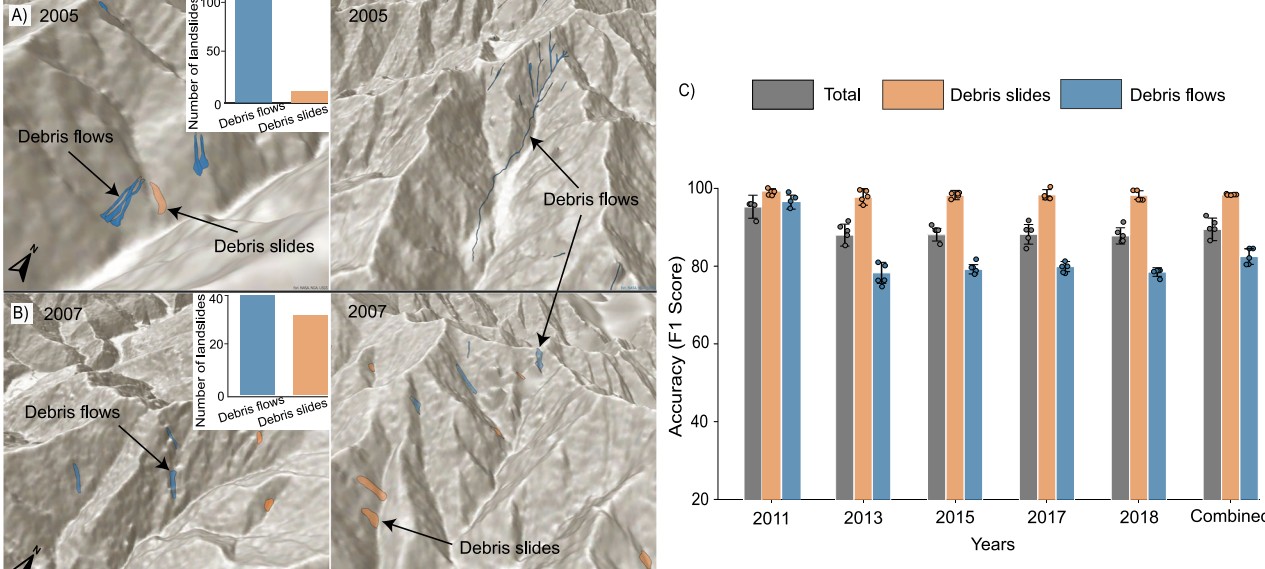

**Fig. 6 | Multi-temporal analysis and prediction on undocumented inventories.** Rows **a** and **b** display snippets of proposed method predictions in two undocumented databases from the 2005 and 2007 inventories of Wenchuan, China, respectively. Each row in these plots shows debris slides (colored in orange) and flows (colored in dark blue) as identified by the model, with the accompanying bar chart quantifying the number of landslides by their type. Plot **c** shows the method performance on multi-temporal inventories triggered by either earthquake or rainfall spanning from 2011 to 2018 on both movement types (also on a combination of them together, colored in gray). The model is trained on the 2008 inventory associated with that year's co-seismic event and tested against each consecutive temporal inventory. The chart also includes a composite dataset derived from combining all the multi-temporal inventories. Error bars represent mean values ± SD (*n* = 5) for each landslide type within each inventory. The model achieved an average F1-score of 89%, with a mean standard deviation (for the F1-score) of ±4%. The base map was sourced from the World Hillshade Map. Map credits: World Shaded Relief-ESRI[68].

scarce or lacking. We highlight the results in the following paragraphs.

Training and testing in regions individually: In this aspect, we test the performance of our method's implementation across diverse geomorphological and climatic settings in the regions of the US Pacific Northwest, Turkey, Denmark, and Wenchuan in China (2008 earthquake event). The evaluation schemes involved training and testing the method individually for each region, where the mean classification accuracy was observed above 80% for each region (see Fig. 5c). This result highlights the adaptability of the proposed method in acclimating to these varying geographical regions.

Implementation in data-scarce contexts: In real-world scenarios, most landslide inventories have limited or no information regarding failure types. To evaluate the method performance in the regions with limited samples, we train and test the method with different amounts of training samples. In Italy, where the landslide samples are distributed around the entire country, even a small subset of samples <1% of entire databases (~100 samples from each class) achieves more than >70% performance. In Wenchuan, China, where landslides are distributed in a basin, with just <20 samples from each class, the method achieves more than 75% performance. These analyses highlight that irrespective of the scale (national- or basin-level), the model can perform well with limited training data. Similar results were also replicated for the US Pacific Northwest, reflecting robustness and applicability across the board. These tests indicate that robust training can be achieved in any region with scarce data. More information can be found in Fig. S3.

Implementation across multi-temporal inventories (temporal transferability): We implemented the method in an event-specific multi-temporal inventory from Wenchuan, China, containing eight event-specific inventories from distinct years, triggered by a combination of rainfall and earthquakes. To assess the method's performance for temporal transferability, we trained the model using the initial earthquake-triggered inventory from 2008. Then we tested it on the remaining inventories from the subsequent years (2011 to 2018). Across all the testing inventories, our method achieved a mean classification accuracy of over 90% (see Fig. 6c), highlighting the temporal transferability of the model. Additionally, we implemented the method on two undocumented inventories predating the 2008 event, specifically from 2005 and 2007 (see Fig. 6a, b). In our prediction of the landslide types within these inventories, we found that most were identified as debris flows (117 in 2005 and 40 in 2007), with the remaining few classified as debris slides (14 in 2005 and 32 in 2007). We manually verified the method's predictions for these landslides using Google Earth archive imagery. Further details regarding the verification of these landslides are provided in Note S5 and Figs. S4 and S5.

## Determining sub-type failure movements

Sub-type classification of landslides is critical for devising targeted mitigation strategies and understanding the underlying mechanisms that drive these movements. To address this need, we go beyond the general categorization of landslide failure types to test the ability of the model to identify/classify specific landslide typologies for the inventories of Italy and the Pacific Northwest of the USA. In the case of Italy, we tested with five sub-type failure movements (slides, debris flows, earthflows, complex landslides, and rock falls) achieving an F1 accuracy of 96% while in the case of the Pacific Northwest of the US, we tested with six sub-type failures (debris flows, earthflows, rotational slides, translational slides, rock falls, and complex landslides) achieving an average F1 accuracy of 84% (see Fig. 5d). This extension demonstrates that our method can handle more than the basic four classes or types, including differentiating sub-types within slides (such as rotational and translational) and flows (including earthflows and debris flows). Such distinctions are vital for type-specific predictive modeling and a comprehensive understanding of landslide movements, as well as the underlying causes of their behaviors. This success can be attributed to the TDA's capability of discriminating among

various failure types, like debris flows and earthflows. This distinction arises from the inherent morphological variations between debris (characterized by more sinuous and longer trails) and earthflows (marked by less sinuous and relatively shorter trails).

### Identifying failure types within the complex landslides

Complex landslides typically occur as amalgamations of numerous processes or failure types that appear successively, such as slides to flows[39]. Because complex landslides include multiple failures, it is challenging to investigate their behavior for the purposes of predictive modeling. Topological properties are capable of capturing intricate information between different failure processes (as seen in the previous sections), and hence, we further explored TDA's capability to understand the underlying coupled failure types that form complex landslides. We utilize 428 complex landslides from the US Pacific Northwest inventory to discern the combination of failure types present in them. Out of 428 complex landslides, 198 of them are documented as "Translational rock slides followed by rock falls" and the rest are documented as "Rotational slides followed by flows" (as reported by the Statewide Landslide Information Database for Oregon, SLIDO[40]).

To identify the failure types within these complex failures, we trained our method with three classes (i.e., slides, flows, and falls) and forced the model to predict the class probability corresponding to each failure type. For 198 complex landslides documented as "Translational rock slides followed by rock falls", our model predicts slide-type failures with the highest probability followed by falls (see Fig. S6d). Similarly, for the remaining 230 "Rotational slides followed by flows" complex landslides, our model predicts slide-type failures with the highest probability followed by flows (see Fig. S6e). Among these 428 landslides, sliding failures are predicted as the most dominant failure type, which is also evident when observing the resemblance of the slide and complex failure topological properties (such as $AL_H$, $BC_H$, and $BA_H$) in the PDF plots (see Fig. S6a–c). These findings demonstrate that topological properties can capture more than just one physical process in a given landslide and that they could be used to automatically improve large Varnes-based[14] inventories toward Cruden and Varnes-based[41] classification.

## Discussion

In this work, we attempted to determine the failure types based on landslide movements through the lens of landslide topology. Our key findings elucidate the connection between the topological properties as a proxy to identify underlying failure movements. We observe that identical landslide types harbor similar topological properties, indicating the presence of common morphological characteristics that govern the general movement of the failures. We find that topological properties offer a more profound capacity to distinguish between failure types than traditional geometric properties (see Results, "Landslide topology as a proxy to identify failure movements"). This finding can be attributed to the fact that topological properties inherently capture critical information related to landslide kinematic progression, failure depth, sinuosity, compactness, and variations in slope. In contrast, geometric properties tend to oversimplify the complex spatial, kinematic, and mechanical relationships that govern the behavior of landslides and are hence less effective in differentiating between various failure types. Building on the advantages of topology over geometry, we developed a method using the Italian landslide inventory that utilizes topology to identify failure types.

The efficacy of our work is showcased through discussions on the method's implementation, the analysis and implications of sub-type movements in landslide risk, deciphering complex landslide interactions, and connecting movement types with process insights. We then pivot to outline our method's inherent limitations for a comprehensive view. The discussion then concludes with an outlook on the method's

potential across geophysical disciplines, underscoring its wide-reaching implications and versatility.

An important step involved recognizing our method's capability to adapt across diverse geomorphological and climatic settings, including both historical (such as Italy, the US Pacific Northwest, Denmark, and Turkey) and event-based inventories (such as Wenchuan, China). This exploration yielded noteworthy results, as the method is capable of identifying failure types across these diverse regions. Thus, the method can help identify failure movements in previously undocumented inventories. Consequently, recent events and their corresponding inventories would benefit from this approach, as they could be efficiently classified, particularly in data-scarce regions (see Results, "Method implementation for failure-type identification").

Beyond adapting to different regions, our approach also opens possibilities for studying the complexity and evolution of landslide behavior, with the potential to improve hazard forecasting[37]. As demonstrated in the temporal experiment in Wenchuan, China, our approach admittedly holds the ability to identify failure movements across not just space but also time, with which we were able to also recognize and quantify the failure movements in two undocumented inventories predating the Wenchuan earthquake event (Mw 7.9) of 2008. Such information carries substantial importance as it paves the way for focused research into quantifying (re-)mobilization of failures, extracting precise data on sediment budgets, and understanding dominant geophysical cycles at continental and global levels[42–45].

Drawing on the experiments of the sub-type classification (e.g., debris flows and earthflows) of landslide failures using the Italian and the US Pacific Northwest dataset (see Results, "Determining sub-type failure movements"), we recognize the substantial value in identifying and quantifying these sub-movements, bearing notable potential to enhance both landslide risk assessment and related hazard models[20]. The level of damage to infrastructure and the risk of human casualties vary depending on the intensity of the failure movement, which differs for each failure type[14]. For example, a slow-moving deep-seated rotational landslide (1.5 m/year to 16 mm/year) may not pose an immediate threat to the population, but it can cause extensive structural damage to buildings over a prolonged period[46,47]. In contrast, flow-type failures, such as debris flows, have rapid mobility and can result in significant casualties and infrastructure damage simultaneously[48,49]. Similarly, episodic impacts in fall-type failures can cause massive damage to infrastructure in a matter of seconds due to their high energy (e.g., impact pressure measured in kilopascals, kPa)[50]. We can infer from these broad examples that the availability of failure-type, especially sub-type, information could considerably enhance the accuracy of predictive modeling and that incorporating it benefits the landslide community as it enables the development of accurate landslide predictive models. Nevertheless, certain failure movements fall in between, for instance, avalanches (flow-type processes) between slides and flows, where avalanches can occur in similar topographical conditions as slides[13]. The presence of larger boulders in avalanches than debris flows has concrete implications for the protective measures required to absorb the impacts. We discuss this facet, particularly, separating the intertwined movements of avalanches from debris slides, in Note S6.

In addition, we utilized topological properties to dive deeper into complex landslides and identify the underlying coupling of failure types that contribute to their formation (see Results, "Identifying failure types within the complex landslides"). Our findings in the US Pacific Northwest suggest that topological properties can reveal more than one physical process in a given landscape (for example, identifying coupled failures of slides following falls or slides following flows). This has significant implications for understanding complex landslide failures, which often arise from a combination of different failure movements, such as sliding, flowing, and falling, and may not be fully

encompassed by conventional characterization and classification methods for large-scale analysis[13,14,41]. Traditional approaches may struggle to pinpoint the exact cause, leading to hindrances in prevention efforts. The use of topological properties to uncover these intricacies offers a path towards a more comprehensive characterization, where capturing the dominant failure type would enable pinpointing the possible initial failure within the complex landslide. Given these advantages, we anticipate that our method will open avenues for future research, particularly in the landslide modeling community, for example, working on large-scale post-event mapping and large- to medium-scale hazard forecasting.

The 3D topological descriptions of landslides do not directly connect landslide failure movement to the failure process, as they rely solely on the topography and the shape of the landslide. In establishing characteristics for failure movements with topological information, the model provides insights into the predominant landslide types prevalent in the landscape. Regardless, it is important to recognize that 3D-based topology alone may not directly lead to the underlying processes driving landslides. Incorporating ancillary information such as land use characteristics (e.g., vegetation or lithology) and information on the triggers (e.g., peak-ground acceleration or rainfall intensity) into future models could tailor the characterization of mapped landslides to assess particular failure processes.

Our approach has the potential to link the movement types that emerge from anthropogenic activities with characteristics that may deviate from naturally induced landslides. For instance, fill-slope landslides induced by human activities might resemble naturally-induced deep-seated landslides in appearance, but their underlying processes are distinctly different. This overlap poses challenges to effectively comment on the processes behind landslide failures (between urban and natural landslides) that look similar. Anthropogenic activities such as land use change, urbanization, and soil management practices have various impacts on landslide activities, sometimes leading to complex scenarios[47]. If trained with appropriate data, our model could effectively distinguish between landslide types in urban and natural environments and be instrumental in studying the disparities in between. By doing so, the method can improve mapping and classification protocols, which may contribute to understanding how human pressures, particularly in urban areas, are influencing the timing and characteristics of landslide failures[51].

While the proposed method demonstrates notable success in identifying failure types, there are inherent limitations. Although the method adapts across regions (US Pacific Northwest, Turkey, Denmark, and Wenchuan in China) and temporal settings (Wenchuan, China), the accuracy of the results is limited by the specific landslide movement types available during training. Hence, all new landslide types must be introduced to the model in advance. We showcased that the model can easily accommodate additional landslide types if desired (see Results, "Determining sub-type failure movements of the landslides", Fig. 5d). We suggest continual updates to both the model and inventories with new movement types as they become available to sustain reliable performance.

Another concern about our approach is transferability. An ideal solution would be a transferable model that can be trained and tested in geographically disparate areas. However, the development of such an ideal solution is exceptionally challenging. Model training and testing in such disparate conditions can introduce bias since distinct failure types from different regions can have dissimilar shapes and thereby, dissimilar topological responses, even for the same failure movements[13]. This challenge is due to diversities in the landscape features, both internally by geology and topography and externally by climate and tectonic activity worldwide, and therefore, impacts how landslides propagate in varying ways in different parts of the world. Alternatively, comprehensive training landslide data that encompasses the diverse landscapes of the earth can aid in achieving a global generalization.

However, such detailed-curated data is unavailable at the moment. More insights into the technical limitations are explored in Note S7.

In real-world scenarios where the landslide databases are undocumented, i.e., devoid of any information regarding failure types, practitioners and experts can train the model on a manually annotated small subset of samples and then classify the remaining landslides in the inventory using our method. The amount of samples that need to be annotated depends on the scale of the region. For example, in Italy, ~100 samples from each class achieved satisfactory results, whereas, in Wenchuan, China, <20 samples from each class were enough. This practical solution ensures an economical direction to obtain reasonable prediction capability in new study areas (see Fig. 5b and Fig. S3). Additionally, we chose the combination of TDA with decision tree-based shallow learning[52] (i.e., random forest) deliberately to capitalize on a functioning model for data-scarce contexts to ensure practicality in the real world. This solution expresses a streamlined approach highlighting the importance of dynamic, adaptable models in improving landslide prediction across diverse regions.

The potential of the proposed method reaches beyond just understanding the complex interplay between landforms, their shapes, and the underlying geophysical processes responsible for their formation; they also serve as a subject captivating interest across various geophysical disciplines. The ability to acquire knowledge about the processes generating complex landforms based solely on their shapes suggests a rich presence of signatures imprinted on the landscapes. Our method leverages their topological properties to effectively extract this information. Envisioning compelling applications beyond landslides, we can explore other geophysical processes such as permafrost-borne retrogressive thaw slumps in Arctic regions[53] and sub-surface processes, e.g., submarine landslides[54], which commonly occur in a typical data-scarce environment such as the sea bottom, where geological and geotechnical information are almost absent. Furthermore, the lens of topological characteristics can enable research on extra-terrestrial landslides such as those on Mars. The topology of Martian rock avalanches, slumps, and slump-flows with cryosphere characteristics[55] can assist in understanding the paleoenvironmental conditions on the planet. Characteristics of landslide mobility could provide insights into material properties and the conditions of sediment deposition at their occurrence time, for example, the presence of water and ice content[55,56]. These geomorphological processes give rise to unique landforms displaying distinct shapes and configurations, both terrestrial and extra-terrestrial, and employing topology can aid in gauging the mechanisms governing their occurrences. By doing so, our method could provide alternate perspectives on mathematical and physical phenomena underlying various geophysical and environmental scenarios.

## Methods
### Topological feature engineering
In the proposed method, landslide polygons serve as the primary input. These polygons represent the 2D outline of the landslide body on the ground and are commonly found in landslide databases. Each vertex of the landslide polygon comprises geographical latitude and longitude coordinates. Utilizing the digital elevation model, the landslide polygons are transformed into normalized 3D shape outlines, wherein each vertex encompasses latitude, longitude, and elevation information. Topological data analysis (TDA) is employed to extract the geometrical and topological characteristics of a landslide's 3D shape outline (see Fig. 7). This information is subsequently used as input for a machine learning algorithm, specifically the random forest. The Python library Giotto-TDA is leveraged to extract an assortment of TDA properties/features from the 3D shape of landslides[57]. To ascertain the most pertinent features for landslide-type classification, a correlation test is conducted between TDA features, and those with high correlation are removed. The remaining, less correlated features

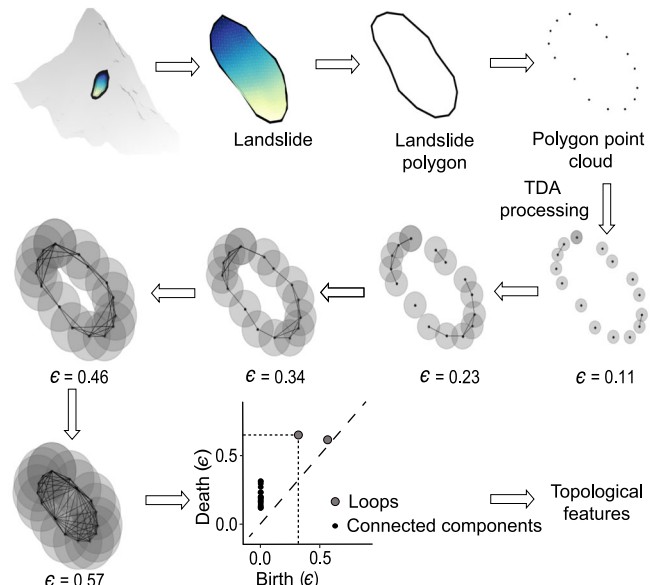

**Fig. 7 | Flowchart of topological data analysis.** The diagram illustrates the procedure of computing topological features corresponding to a landslide shape. The color gradient in the example landslide depicts relative elevation from the crown (colored in dark blue) to the toe (colored in light green). The flowchart shows the use of persistent homology in capturing various structures of the landslide shape by using an evolving disk size ($\epsilon$) around each point in the point cloud. With the increase in $\epsilon$, various structures like connected components and holes emerge in the data's shape which is captured by the persistence diagram. Using this information, we can calculate the topological properties of the landslide's shape. Please note that when processing the topological data analysis (TDA) features, we display the flowchart using a 2D illustration for simplicity and better visualization.

are then assessed, and the least important ones are iteratively eliminated until six robust predictors remain. The exclusion of additional predictors results in decreased performance, while incorporating more than seven yields comparable outcomes. Utilizing fewer predictors facilitates the development of a more generalizable model. The six features thus form a feature space for the random forest classifier, which is then evaluated using a confusion matrix and other accuracy metrics like True Positive Rate (see Note S8 and Fig. S7).

Topological data analysis (TDA) quantifies the multidimensional shape of data using algebraic topology techniques. TDA offers a variety of metrics for capturing the geometric and topological properties of data shapes[58]. These metrics could be used as a feature space for the machine learning algorithm to solve various classification and regression problems, such as shape classification. TDA's central idea is persistent homology, which identifies persistent geometric features by using simplicial complexes to extract topological features from point cloud data. Simplicial complexes are a collection of simplexes that are the building blocks of higher-dimensional counterparts of a graph. An n-dimensional simplex is formed by connecting $n+1$ affinely independent points[59,60]. For example, a point is a 0-dimensional simplex, an edge that connects two points is a 1-dimensional simplex, and a filled triangle formed by combining three non-linear points is a 2-dimensional simplex. A Vietoris-Rips complex indicates the simplicial complex in the data's shape using a parameter $\epsilon$. The main idea of the Vietoris-Rips complex is to connect any two points in the point cloud data set whose distance is less than $\epsilon$. These connections of data points create structures in the data that change with the parameter $\epsilon$. Therefore, to get complete information about all the structures in the data, the idea is to use all $\epsilon > 0$ values.

Only specific structures in the data shape provide crucial information about the geometrical and topological properties of the data. Homology measures these unique structures in the data, where e.g.,

0-dimensional homology captures connected components or clusters, 1-dimensional homology measures loops, and 2-dimensional homology measures voids[59]. These crucial structures emerge and die with changes in $\epsilon$, and this information is captured in the persistence diagram. With the help of a persistence diagram, we can calculate various measures quantifying the topological properties of the shape—persistence entropy, average lifetime, number of points, Betti curve-based measure, persistence landscape curve-based measure, Wasserstein amplitude, Bottleneck amplitude, Heat kernel-based measure, and landscape image-based measure[61,62]. We have explained all the topological features in detail in Note S3. Finally, we used all these measures as input in the machine learning method–random forest.

## Machine learning model: Random Forest

Random forest is an ensemble-based learning method that has shown promising results in various classification and regression problems[63,64]. Random forest classifiers consist of multiple classifiers trained independently on bootstrapping training samples. Bootstrapping $N$ training samples leads to $\frac{2N}{3}$ independent samples, so each tree in the random forest is constructed from a distinct subset of training samples[65]. Moreover, each tree in the random forest predicts the output class of the testing sample independently, and the class with the majority votes is the final decision of the random forest[65].

Each random forest tree divides a parent node into two daughter nodes, right ($r$) and left ($l$). For each node split, the random forest chooses $p$ features from the $m$ total features of the samples[66]. Among $p$ features, the random forest selects a single feature for a node split based on the "Gini-index" criterion. The Gini-index for each right and left daughter node can be calculated as: $G_r = 1 - \Sigma_{j=1}^{j=N} P_{rj}$ and $G_l = 1 - \Sigma_{j=1}^{j=N} P_{lj}$. Here, $P_{rj}(P_{rlj})$ and $N$ are the probability of the samples in the right (left)nodes having class $j$ and the total number of the classes. The features that maximize the change in the Gini-index that is calculated as follows: $\Delta\theta(s_q) = G_q - \rho_{rq}G_r - \rho_{lq}G_l$ is used for the node split[67]. Here, $\rho_{rq}$ and $\rho_{lq}$ are the proportion of samples in the right and left daughter nodes. The process of splitting nodes continues until a stopping criterion is met, such as when no more samples are available for splitting or when the Gini-index of parent nodes is lower than that of daughter nodes.

## Data availability

Sample data for feature engineering and model training-testing are provided in a GitHub repository[69]. The predicted data generated for the Wenchuan, China 2005 and 2007 landslides inventories as part of the temporal transferability experiment have also been deposited in the same repository.

The dataset utilized for Italy in this study was obtained from the Inventario dei Fenomeni Franosi (Inventory of Landslide Phenomena) in Italy (IFFI)[32]. The IFFI project catalog (www.progettoiffi.isprambiente.it) was created in 1999, with the aim of mapping and identifying landslides in Italy, and holds information on over 250,000 usable landslide polygons. Aerial image interpretation, historical sources, and field surveys were used to acquire and validate this catalog, while the classification protocol or scheme referred to that of Varnes[14] and Cruden and Varnes[41]. In our work, we chose the polygonal landslide data from this catalog and also carried out post-processing to correspond to the spatial extent and resolution of the 25-m EU-DEM[70].

The dataset from the US Pacific Northwest consists of inventories from the Oregon Statewide Landslide Information Database (SLIDO—updated 10/29/2021; Franczyk et al.[40]), mapped by the Oregon Department of Geology and Mineral Industries (DOGAMI), and the Washington State Landslide Inventory Database (WASLID updated 2018/08/01; Slaughter et al.[71]), mapped by the Department of Natural Resources, Washington Geological Survey (WGS). The combined inventories comprise 47,653 landslides from the US Pacific Northwest region. The inventories contain LiDAR-derived landslide polygons

guided by protocol to capture the movement types with spatial information on the scarps, head scarps, toes, and deposits[71-73]. Since this data is categorized using a combination of Cruden and Varnes[41] and Hungr et al.[13] (i.e., slides, flows, complex, and falls), we modified the Italian data correspondingly to maintain uniformity in the taxonomy of the failure mechanisms.

The national landslide inventory of Denmark was obtained from Luetzenburg et al.[74], published in 2022. The landslides were mapped via high-resolution DEM of 2015 and orthophotos supplied by the Danish Agency for Data Supply and Infrastructure, consisting of 3202 unique polygons of mapped landslides following the classification from Hungr et al.[13].

The landslide inventory of Turkey was obtained from Gorum[75] published in 2019. The landslides were mapped from airborne LiDAR data, with a total count of 900 landslides classified according to the Cruden and Varnes[41] system.

Landslide inventory for the Wenchuan region of China was acquired from Fan et al.[76] where they generated a multi-temporal inventory of the infamous Wenchuan 2008 earthquake event causing ~10,000 landslides after the event. The multi-temporal window spans from 2005 to 2018, mapped with a custom classification system based upon a simplification of Hungr et al.[13].

The EU-DEM for Italy and Denmark was downloaded from https://www.opentopodata.org/datasets/eudem/ and the DEM for the US Pacific Northwest was downloaded from https://www.opentopography.org/. The Shuttle Radar Topography Mission (SRTM) DEM for Wenchuan, China, and Turkey was downloaded from https://dwtkns.com/srtm30m/.

## Code availability
Data analysis and processing were conducted using the Python programming language and its associated libraries. The various scripts used for data analysis are available at the GitHub repository[69].

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

## Acknowledgements

The authors extend their gratitude to Dr. Tolga Gorum and Dr. Hakan Tanyas for providing access to the Turkish landslide inventory dataset, which served as an essential resource for further evaluation of our model. K.B. and F. Catani acknowledge that their contribution to the present work was undertaken as part of the "The Geosciences for Sustainable Development" [CUP C93C23002690001] project of the Department of Geosciences, University of Padova. K.R., J.V.F., and F. Cotton acknowledge funding from the NatRiskChange Research Training Group (Deutsche Forschungsgemeinschaft; GRK2043/3). UO acknowledges funding from the research focus point "Earth and Environmental Systems" of the University of Potsdam. K.R. and N.M. acknowledge support from the College of Science, the Chester F. Carlson Center for Imaging Science, and the School of Mathematics and Statistics at Rochester Institute of Technology.

## Author contributions

K.B., K.R., J.V.F., U.O., F. Catani, and N.M. contributed to the conceptualization and design of the research. K.B. and J.V.F. curated the data. K.B. and K.R. developed the methodology and conducted the formal analysis under the supervision of U.O., F. Catani, F. Cotton, and

N.M. U.O. provided day-to-day supervision with writing and reviewing. All authors contributed to writing, reviewing, and editing.

## Competing interests

The authors declare no competing interests.
