## [Peer Review File · Nature Communications]

Landslide topology uncovers failure movementsReviewers' Comments:

Reviewer #1 (Remarks to the Author):

This study explores the use of not only geometric but topologic parameters to identify the types of landslides using their polygon and underlying topographic data. The authors propose a solid workflow for the topological analysis and demonstrates the performance of the automatic classification of landslide types with reasonable accuracy. Other than the cases reported (US Northwest and Italy), the work is potentially useful for further applications in the different settings of the world.

However, there are some concerns regarding the types of landslide failures proposed in this manuscript. The authors classified the landslide types into three (and one additional as a combined type), but the differentiations of those three types are somewhat vague. For instance, the authors do not distinguish shallow and deep-seated "slides" (type 1). Although cohesion is an important factor (as noted by the authors), the size of slides, from small to extra large (such as Seimareh slide, as mentioned by the authors), does affect the actual mechanisms including kinematics. This also applies to "flows", for which materials (typically rock debris or mud) play important roles in their flow mechanisms. "Falls", perhaps mostly dry rockfalls, could be triggered by just gravity and/or earthquakes and rainfalls, but such triggering factors cannot be identified with these classifications. Although the authors mentioned that "Understanding the failure type of a landslide can provide insight into its triggering mechanism", it is hard to say how the identified types correlate with triggering mechanisms. It sounds a bit suspicious whether the proposed method can "uncover failure mechanisms" of various landslides.

It is also unclear how the landslide boundaries were defined, particularly for "falls". The source and deposits of rockfalls (and some debris flows) may be discrete, so that the single polygon may not fit well to cover those areas. This issue needs to be solved carefully because such landslide shapes may differ from place to place according to the landform conditions (steepness), lithological materials, and climatic settings (in humid regions the landslide deposits may be deformed easily).

As already noted by the authors (line 135), the exact physical mechanisms of landslides may not be clear with these classifications. Although the proposed types may be well identified by the topological characteristics, the proposed objectives of this study (to identify landslide mechanisms by topological features) may not be well achieved.

I would therefore recommend revising the manuscript to set an alternative goal (not mechanisms but types), or to increase the sample sites with more various topographic/geologic/tectonic/climatic conditions.

Reviewer #2 (Remarks to the Author):

The manuscript aims to identify landslide failure mechanisms using 3D landslide topology and machine learning techniques. It is an interesting paper with detailed discussions on how topological properties can be effectively used to distinguish between various landslide failure types.

- Landslide types in Figure 1: The various types of landslides can be differentiated by the mode of movement and the kinds of material involved. The authors seem to define landslide types in Figure 1 by only the mode of movement, but ignore the material types, such as clays, sensitive clays and rocks.

Therefore, some landslides were excluded, e.g. lateral spreading of clay/sensitive clays. Moreover, debris flows triggered by rainfall (very common in Italy) and erosion-triggered mud flows are very different. It is doubtful that these two types of landslides can be merged to 'Flow' type. Please explain why only four types of landslides were defined and analyzed in this study.

- Line 42, Page 2: the statement 'For instance, they may predict a low probability of landslide occurrence in a high landslide-prone region' is subjective. Please add more details to explain.

- Figure 4: the sum of the percentage values in Figures 4(a) – (f) isn't equal to 1. It is confusing that BCC is more important than BCH, WAC and BAH indicated by Figures 4(c) to (f), while BCC is less important in Figure 4g. Please explain.

- Lines 187-192, Page 8: model transferability is an issue in this study. It may be difficult to generalize the ML model to be used in different regions. It is usually not possible to obtain a large dataset in the regions of interest and thereby not possible to make a model using the methods proposed in this study. Thus what's the main contribution of this study?

Reviewer #3 (Remarks to the Author):

Key results:

Despite advancements in predictive modeling, the death toll and monetary damages caused by landslides are still increasing. The Authors argue that this is because of the limited predictive capability of existing models, which often need more crucial information about the underlying landslide failure mechanisms. To address this issue, the authors propose a new approach that leverages 3D landslide topology to identify failure mechanisms, including flows, falls, slides, and complex movements. By analyzing topological proxies, they discover distinct patterns that reveal the mechanics of mass movement embedded in the shape of the landslides. They find that landslides with identical failure mechanisms exhibit similar topological properties, allowing them to accurately identify failure mechanisms in Italy and the US's Pacific Northwest region with an accuracy ranging from 80% to 94%. These findings can significantly enhance the performance of landslide predictive models and impact assessments. Additionally, this work introduces a new paradigm for studying landslide shapes and understanding the underlying processes through the lens of landslide topology.

Validity:

How critical is the scale at which a landslide polygon is mapped? Are the landslide polygons mapped

consistently (e.g., 1:5,000 or 1:10,000)? Do results depend upon how many nodes (digitization points) a mapper includes for a landslide polygon? What if the landslide polygons were mapped with bezier curves rather than straight line segments? Would this complicate or obfuscate the analysis? I guess as a reader, I would have liked to see an example where the authors took a landslide dataset for which there is no information on failure mechanism, applied their model, and told the reader how many of the landslides in the inventory are slides, flows, complex, or fall-types, and what this information says about the triggering mechanisms. This is the missing link to me in the presentation. I would like to see that they can differentiate seismically-triggered versus rainfall-triggered landslides. For example, there are several outstanding datasets that they could work with - the 2022 Haiti earthquake that post-dated the arrival of a tropical cyclone by several days or a decade and a half since the 2008 Wenchuan earthquake. In both cases, some landslides in a combined inventory are derived from rainfall triggering, others from the primary earthquake. Can they differentiate between the two and partition different landslide typologies to different triggering mechanisms and how this might evolve through time (e.g., for the hillslopes affected by the Wenchuan event)? I did not see an example of how the authors would use their results (e.g., Figure 4a-f or Fig. S3a) to create a more informed landslide inventory that still needed failure mechanism results (estimates).

Significance:

This is an interesting study, and I see its potential, even if not fully demonstrated in this manuscript. I wonder if this manuscript would be better suited to a discipline-specific journal, such as Landslides or Natural Hazards, rather than Nature Communications. I'm just not convinced that the observation that landslides with different failure mechanisms have different typologies is a game-changer. Perhaps, if the authors had provided an example where they used the results, as presented in Fig. 4a-f and Fig. S3a, to perform an analysis of a varied landslide inventory from a paleo-event without failure mechanism information, was successful. I would be more confident of the significance/utility of this study.

Data and methodology:

The approach appears valid, although admittedly, I am no expert on topology as applied to 2 or 3-D datasets. The two landslide inventories used, the one from Italy and the other from the Pacific Northwest (Oregon and Washington), are certainly two of the larger datasets available and likely of high overall quality. I know it may not be standard in Nature articles to put the Methods section before the results, but I had a hard time interpreting the results until I skipped ahead to the Methods section. Consider placing the methods before the results and discussion. It makes it so much easier to understand and interpret the results.

Analytical Approach:

The analytical approach used in the investigation is robust and comprehensive.

Suggested Improvements:

Line 19: Reference 1 - This article concerns landslide impacts in Germany. Why is it suitable for a statement about landslide damages worldwide?

Line 22-23: Reference to global annual landslide volume of "fifty-six billion cubic kilometers." Is this the

correct number? This would be the equivalent volume of filling up the Grand Canyon in the U.S. 13,440,000 times with landslide debris yearly (the Grand Canyon has a volume of 4,166 km³. Put another way; this reported volume of material would fill up the Caspian Sea (716,112,531 times EVERY YEAR)! This can not be correct. Do the authors mean fifty-six billion cubic meters, perhaps?

Line 31: What is meant by the phrase “baffling predictive models?” Do you mean that the predictive models are baffling? As in, it is “baffling” that anyone uses the predictive model, or do you mean that the lack of information about failure mechanisms of mapped landslides baffles predictive models? Models are only as good as the data fed into them, so the phrase “baffling predictive models” confuses me as a native English-speaking reader.

Lines 42-43. I do not follow the logic that suggests that because predictive models may indicate a low probability of landslide occurrence in a high landslide-prone region; therefore, the information needed to improve the model is identifying landslide failure mechanisms. The models could fail for many reasons. Garbage in = garbage out when it comes to models. I’m not convinced that the model fails because of a lack of landslide failure mechanism data rather than, for example, poor topographic or geologic data at the regional to global scale.

Line 50-51: “Debris deposits at the talus” does not make sense. Talus is a deposit of rock debris at the base of a slope. The AGI Glossary definition of “talus” is Rock fragments of any size or shape (usually coarse and angular) derived from and lying at the base of a cliff or very steep, rocky slope.

Line 87-88: “...deposited at the talus” does not make sense, especially for flow-type deposits, which usually are deposited on a debris fan (do you mean talus cone or fan?)

Line 104: Fig. S1 -flows do not “follow the contours of the landscape. Rather, they cross contours, where contours represent lines of equal land-surface elevation. A more correct phrase would be “flows are the most sinuous, following the channelized topography more closely than other mass-wasting mechanisms.”

Line 204: What do you mean by “manually annotate a few instances to train the model?” By “few,” do you mean 500? Still, quite a few to manually annotate.

Figures:

Figure 1: For Type 1, “Talus” is misleading, as it does not need to be rock, which is what talus is composed of. “Debris” would be a better term.

Also, for Type 1, “Scarp” is a landform, whereas “deposition” is a process. Thus you are mixing a thing (landform) with a process. How about “erosion” and “deposition,” and then if you want to label the scarp, move it to the right side of the topographic profile (like you have for the label “talus.”) Similar comment for the Type 2 Cross Section. How about changing “Scarp” to “Evacuation?” You already have “Scarp (at regolith)” on the left side of the topographic profile. No need to have it twice on the same diagram.

In the caption, consider replacing the highlighted instance of “flows” with “moves” or “travels” since you already use the word “flow” two other times in the same sentence.

What are the little blue symbols on the 3-D diagram next to Type 2 and 4 landslides? Are these rain clouds, lakes, or something else? Why show the houses and trees? Do the landslides impact them? Are they important to the different failure mechanisms highlighted in Figure 1?

What does the dashed line just below the top of the “Steep slopes” polygon indicate? What is this important? Is the dashed line the steep slope break, or is the red polygon the area with steep slopes? Do you need both on the figure? Your Type 2 flow-type failure starts where slopes are not steep on the 3-D diagram.

Figure 3:

Part A's caption reads, “The diagram also shows the 3D landslide polygon, which outlines the landslide shape.” Isn't this a 2D polygon, the one on the right with the white center and black outline? I don't see any 3D information associated with this part of the figure.

References:

Yes. The manuscript references previous literature appropriately.

Line 398: What is the publication year for reference #21?

Supplemental Information (SI):

S2. First Paragraph. “Accumulated debris at the talus” does not make sense. Talus is accumulated debris (mostly rock) at the base of a steep slope.

Fig. S1 - Caption: Flow-type failures [all failures] do not “follow the landscape's contour” they cross contours, where contours represent lines of equal land-surface elevation.

Clarity and context:

Yes, as long as the reader has a basic understanding of landslide processes and landslide morphologies and the use of topology to investigate the 2D and 3D geometries of polygons.

Your expertise:

I have extensive knowledge of landslides, mass-wasting processes, and the creation and curation of landslide inventories, especially from the Pacific Northwest region of the United States. I have minimal knowledge regarding typology as applied to landslide studies, although I understand the utility and quantitative appeal.

Author's response to comments

September 5, 2023

We thank the reviewers for acknowledging our work, taking the time to read it, and providing suggestions to improve the work. We hope the reviewers will find our revised paper better suited for publication. Below are the point-by-point responses to the reviewer's comments.

Response to reviewer 1

This study explores the use of not only geometric but topologic parameters to identify the types of landslides using their polygon and underlying topographic data. The authors propose a solid workflow for the topological analysis and demonstrates the performance of the automatic classification of landslide types with reasonable accuracy. Other than the cases reported (US, Northwest, and Italy), the work is potentially useful for further applications in the different settings of the world.

Reply: We are sincerely grateful for the reviewer's constructive and encouraging review of our study. We appreciate the reviewer's recognition of the robustness of our workflow.

However, there are some concerns regarding the types of landslide failures proposed in this manuscript. The authors classified the landslide types into three (and one additional as a combined type), but the differentiations of those three types are somewhat vague. For instance, the authors do not distinguish shallow and deep-seated "slides" (type 1). Although cohesion is an important factor (as noted by the authors), the size of slides, from small to extra large (such as Seimareh slide, as mentioned by the authors), does affect the actual mechanisms including kinematics. This also applies to "flows", for which materials (typically rock debris or mud) play important roles in their flow mechanisms. "Falls", perhaps mostly dry rockfalls, could be triggered by just gravity and/or earthquakes and rainfalls, but such triggering factors cannot be identified with these classifications.

Reply: We thank the reviewer for this important comment, which has significantly improved our manuscript. In the revised manuscript, we added a new subsection in the Result section where we showed the method's capability

in identifying landslide sub-types (e.g., debris flows, rotational slides, etc.) in Italy and the US Pacific Northwest. We refer the reviewer to Figure 5-d where for both Italy and the US data, we are able to discern the sub-failure types with very high accuracies.

The reason we first wanted to use just 4 types was because the typologies of slides, flows, falls, and complex are some of the most general ones that we can find in any given context before really examining them into further sub-classifications. The classification of landslide types varies by country, institution and expert opinion, with some inventories following systems like Cruden and Varnes, Hungr, Hutchinson, or a general description of the observed failure movements.

Although the authors mentioned that "Understanding the failure type of a landslide can provide insight into its triggering mechanism", it is hard to say how the identified types correlate with triggering mechanisms. It sounds a bit suspicious whether the proposed method can "uncover failure mechanisms" of various landslides. It sounds a bit suspicious whether the proposed method can "uncover failure mechanisms" of various landslides.

Reply: We thank the reviewer for pointing out the problem in this statement. We agree that just knowing the failure types doesn't provide information about their trigger. For example, a landslide database containing primarily flows can be associated with either earthquake or rainfall triggers. So, we removed this sentence from the revised manuscript to avoid any ambiguity.

This also made us realize the mistake we made in using the terms 'landslide type' and 'landslide mechanism' interchangeably, which was made from a conceptual point of view to express landslide failure types (based on the mode of movement; sliding, flowing, falling etc.). In hindsight, we realize that this was not the best course of action as it introduces ambiguity when talking about the exact mechanisms of the sliding, flowing, and falling behavior, which generally involves information about soil permeability, cohesion, pore water pressure, etc. Consequently, we understand that the aforementioned statement oversimplified the relationship between landslide types and their triggering mechanisms, and we appreciate the reviewer for pointing this out. Therefore, we have amended the term 'mechanism' to 'types' in the entire manuscript, including the title of the manuscript.

It is also unclear how the landslide boundaries were defined, particularly for "falls". The source and deposits of rockfalls (and some debris flows) may be discrete, so the single polygon may not fit well to cover those areas. This issue needs to be solved carefully because such landslide shapes may differ from place to place according to the landform conditions (steepness), lithological materials, and climatic settings (in humid regions the landslide deposits may be deformed easily).

Reply: We appreciate the reviewer's insight on this matter, as it aligns with our own concerns. The generation of fall-type boundaries is indeed significant. While single polygons might not perfectly encapsulate rockfalls, our model mitigates this by considering slope variation within these single polygons, as detailed in the first sub-section of the Results section ('Landslide topology as a proxy

to identify failure types based on style of movement'). The topological parameter Betti Curve of connected components captures variations in slope steepness along the z-dimension, thereby capturing the inherent morphology of the fall types. As such, our model can distinguish fall types from other landslide typologies based on these variations. However, it's important to note that we rely on nationwide institutions for these inventories (such as in the case of Italy and the US Pacific Northwest), comprising thousands of rockfall polygons. The methods employed to demarcate these boundaries may vary and are beyond our control. Achieving homogeneity across all rockfall inventory boundaries, which encompass approximately 20,000+ landslides, poses a challenge. Nevertheless, when mapping encompasses scarp regions or the entire landslide trail, the topological parameter Betti Curve of connected components (BCC) can recognize changes in slope gradient to identify fall-type failures. Thus, our method effectively leverages available data (slope steepness) to model these fall-type failures, and this can be seen in the results for the landslide inventories in Italy, the US Pacific Northwest, Denmark, and Turkey.

As already noted by the authors (line 135), the exact physical mechanisms of landslides may not be clear with these classifications. Although the proposed types may be well identified by the topological characteristics, the proposed objectives of this study (to identify landslide mechanisms by topological features) may not be well achieved. I would therefore recommend revising the manuscript to set an alternative goal (not mechanisms but types), or to increase the sample sites with more various topographic/geologic/tectonic/climatic conditions.

Reply: We appreciate the Reviewer's feedback and acknowledge the terminological confusion between 'type' and 'mechanism'. In our original manuscript, these terms were conceptually interchanged to relate landslide types, such as rockfalls, slides, and debris flows, to their respective mechanisms like falling, sliding along rupture surfaces, and fluidization of material. We recognize that this may have led to some ambiguity. With this understanding, we have refined our manuscript to ensure accurate usage of these terms. We clearly differentiate between landslide 'types', which describe movement patterns, and 'mechanisms', which elucidate the underlying triggers and dynamics of these movements.

We have also added three more study areas (Denmark, China, and Turkey) to showcase the working of our method in other sites, while also adding a new section that showcases the application of the method on spatially and temporally distinct event inventories (multi-temporal inventory of China from 2005 till 2018)(Please see the transferability result sub-section).

Reviewer 2

The manuscript aims to identify landslide failure mechanisms using 3D landslide topology and machine learning techniques. It is an interesting paper with detailed discussions on how topological properties can be effectively used to distinguish between various landslide failure types.

Reply: We are truly grateful for the reviewer’s encouraging comments on our manuscript.

- Landslide types in Figure 1: The various types of landslides can be differentiated by the mode of movement and the kinds of material involved. The authors seems to define landslide types in Figure 1 by only the mode of movement, but ignore the material types, such as clays, sensitive clays and rocks. Therefore, some landslides were excluded, e.g. lateral spreading of clay/sensitive clays. Moreover, debris flows triggered by rainfall (very common in Italy) and erosion-triggered mud flows are very different. It is doubtful that these two types of landslides can be merged to ‘Flow’ type. Please explain why only four types of landslides were defined and analyzed in this study.

Reply: We thank the reviewer for this important comment, which has significantly improved our manuscript. In the revised manuscript, we showed the method’s capability to identify more than four types (for example, debris flows and earth flow) in a new subsection in the Result section. We refer the reviewer to Figure 5-d where using both Italy and the US data, we discern the sub-failure types with very high accuracies. While we acknowledge the reviewer’s point about the importance of considering material types like clays, sensitive clays, and rocks in failure identification, obtaining such detailed data—whether through lithological studies or field visits—is often logistically challenging on a global scale. Moreover, we recognize that additional sub-types may exist, but the limitation of our current dataset restricts our capacity to identify them. Both Italy and the US Pacific Northwest only had the respective sub-types that we used in Figure 5-d of a reasonable sample size. Additionally, the sub-types illustrated in Figure 5-d were selected because they were the only ones with a reasonable sample size available in both Italy and the US Pacific Northwest.

Finally, to address the question regarding the choice of only 4 types, the classification of landslide types can vary by country, institution, and expert opinion, with some inventories following systems like Cruden and Varnes, Hungr, Hutchinson, or a general description of the observed failure movements. To address this inconsistency, we focused on homogenizing the most easily discernible parent failures—slides, flows, falls, and complex— in a simplistic manner while acknowledging the importance of detailed sub-classes for enhancing predictive models. Hence, for simplicity, we show only four failure types in Figure 1 (because of their broad definition of failure movements), despite the ability to capture more than four types of movements.

- Line 42, Page 2: the statement ‘For instance, they may predict a low probability of landslide occurrence in a high landslide-prone region’ is subjective. Please add more details to explain.

Reply: We thank the reviewer for highlighting the subjectivity in this statement. The performance of landslide predictive models depends on topographic, geological, climatic, failure type, and other crucial information. Having incorrect information about one or more input parameters of predictive models will create bias in the models. So we expect that incorrect failure type will lead to bias in predictive models and that might lead to incorrect predictions. We used this line of logic when writing this statement. However, since the effect

of one input parameter on a predictive model is not clear or evident in past studies, this statement creates ambiguity and confusion. We thank the reviewer for pointing this out and we have decided to remove this particular statement from the revised manuscript to avoid this confusion.

- Figure 4: the sum of the percentage values in Figures 4(a) – (f) isn't equal to 1. It is confusing that BCC is more important than BCH, WAC and BAH indicated by Figures 4(c) to (f), while BCC is less important in Figure 4g. Please explain.

Reply: We appreciate the reviewer's attention to detail in noticing that the total sum exceeds 1 (101%). This discrepancy is due to the rounding-up of values for each topological property, which collectively pushes the total sum over 1. However, when using the exact values, the sum is indeed equal to 1. The feature importance of a feature depends on the multi-collinearity in relationship to all the other features. In the context of Figures 4 (a)-(f), BCC has higher importance due to the innate relationship amongst the topological features where we examined colinearity. However, in Figure 4(g), the test was conducted between topological and geometrical feature sets where the geometrical features of eccentricity and convex hull measure presented higher relative importance than BCC.

- Lines 187-192, Page 8: model transferability is an issue in this study. It may be difficult to generalize the ML model to be used in different regions. It is usually not possible to obtain a large datasets in the regions of interest and thereby not possible to make a model using the methods proposed in this study. Thus what's the main contribution of this study?

Reply: We thank the reviewer for raising an important issue about the transferability of the method. Based on the reviewer's comment, we showed the transferability of the approach in various geographic and climatic settings: Denmark, the USA, Turkey, and China. In each scenarios method achieved above average 80% performance. Moreover, We implemented the method in two undocumented databases from China and manually verified these landslide types using ArcGIS and Google Earth archive imagery (Please see Figure 5-a and b and the Results sub-section on transferability for more information).

Moreover, we show that our method doesn't need a large dataset in the region of interest, and with just 100 samples, the method achieves more than 70 percent performance (Please see results subsection on transferability, figure 5-b). For areas entirely lacking documented failure information, the model can be used by manually annotating a modest set of around 100 samples. This provides a practical solution for a streamlined, automated failure-type classification system in documenting failure types in landslide databases, often comprising thousands of samples.

Reviewer 3

key results: Despite advancements in predictive modeling, the death toll and

monetary damages caused by landslides are still increasing. The Authors argue that this is because of the limited predictive capability of existing models, which often need more crucial information about the underlying landslide failure mechanisms. To address this issue, the authors propose a new approach that leverages 3D landslide topology to identify failure mechanisms, including flows, falls, slides, and complex movements. By analyzing topological proxies, they discover distinct patterns that reveal the mechanics of mass movement embedded in the shape of the landslides. They find that landslides with identical failure mechanisms exhibit similar topological properties, allowing them to accurately identify failure mechanisms in Italy and the US's Pacific Northwest region with an accuracy ranging from 80% to 94%. These findings can significantly enhance the performance of landslide predictive models and impact assessments. Additionally, this work introduces a new paradigm for studying landslide shapes and understanding the underlying processes through the lens of landslide topology.

Reply: We thank the reviewer for their in-depth reading of our manuscript and for providing us with valuable comments and insights to improve this manuscript.

How critical is the scale at which a landslide polygon is mapped? Are the landslide polygons mapped consistently (e.g., 1:5,000 or 1:10,000)? Do results depend upon how many nodes (digitization points) a mapper includes for a landslide polygon? What if the landslide polygons were mapped with bezier curves rather than straight line segments? Would this complicate or obfuscate the analysis?

Reply: We thank the reviewer for asking this interesting and important question regarding the landslide polygon quality. Our method uses TDA—a topology-based method to capture properties of data's shape (landslide's shape). TDA has been found to be resistant to noise in the data's shape and captures the global properties of a given shape rather than local properties. Our method achieved above 90 percent performance in historical inventories (e.g., Italy where generally, inventory for each province is developed by its own municipality/civil protection agencies/universities) including landslides mapped using various mapping scales. So we believe that mapping scale or number of nodes will not affect the performance of the method in identifying failure types.

I guess as a reader, I would have liked to see an example where the authors took a landslide dataset for which there is no information on failure mechanism, applied their model, and told the reader how many of the landslides in the inventory are slides, flows, complex, or fall-types, and what this information says about the triggering mechanisms. This is the missing link to me in the presentation. I would like to see that they can differentiate seismically-triggered versus rainfall-triggered landslides. For example, there are several outstanding datasets that they could work with - the 2022 Haiti earthquake that post-dated the arrival of a tropical cyclone by several days or a decade and a half since the 2008 Wenchuan earthquake. In both cases, some landslides in a combined inventory are derived from rainfall triggering, others from the primary earthquake. Can they differentiate between the two and partition different landslide typologies to different triggering mechanisms and how this might evolve through time

(e.g., for the hillslopes affected by the Wenchuan event)? I did not see an example of how the authors would use their results (e.g., Figure 4a-f or Fig. S3a) to create a more informed landslide inventory that still needed failure mechanism results (estimates).

Reply: We appreciate the reviewer’s comments as it helped us in showing a real-world application of our work and also laid the foundation for future direction. Based on the suggestions, we implemented our method on two undocumented paleo-event dataset obtained from the multi-temporal landslide inventory of Fan et al. (2019) covering the years between 2005 and 2018. We identified the landslide failure movements for the undocumented inventories of 2005 and 2007. Most of the landslides were recognized as debris flows followed by debris slides, for which we manually verified these landslides using ArcGIS and Google Earth archive imagery (Please see Figure 5-a and b, and the Results sub-section on transferability for more information). However, we cannot conclusively comment on the associated triggers of these two inventories solely based on the types. Although the predominant failure movements are identified as debris flows, a clear connection between this failure to a trigger cannot be made directly as it may be triggered either by rainfall or seismic activity. We release these two now-documented inventories to the public via the corresponding author’s GitHub (link: <https://github.com/kushanavbhuyan/Uncovering-landslide-failure-types>).

Moreover, we show the method’s capability in identifying failure in event-specific inventories by training the model on the 2008 event inventory and predicting failure types temporally from 2011 to 2018, achieving over 80% classification accuracy on average.

At the same time, this puts forward an interesting research question: Do distinct failure types with different trigger mechanisms have dissimilar topologies? For example, are there any dissimilarities between slide-type failures triggered by rainfall and by earthquakes or flow-type failures triggered by rainfall and earthquakes? Based on this intriguing insight, we plan to continue our research in this direction in the future as it could help establish a statistically-based topological connection between trigger types and failure movements. However, at this point, we skip this as it is out of scope and also time and labor-intensive to find adequate databases to develop this idea for linking failure types to trigger mechanisms.

This is an interesting study, and I see its potential, even if not fully demonstrated in this manuscript. I wonder if this manuscript would be better suited to a discipline-specific journal, such as Landslides or Natural Hazards, rather than Nature Communications. I’m just not convinced that the observation that landslides with different failure mechanisms have different typologies is a game-changer. Perhaps, if the authors had provided an example where they used the results, as presented in Fig. 4a-f and Fig. S3a, to perform an analysis of a varied landslide inventory from a paleo-event without failure mechanism information, was successful. I would be more confident of the significance/utility of this study.

Reply: We appreciate the reviewer’s suggestion and comment. To address this concern, we demonstrate the applicability of our method in two undocumented databases. Please see the results of the 2005 and 2007 (paleo-event inventories) predictions of landslide types in Wenchuan, China (Figure 5-a and b, and the Results sub-section on transferability), which we also release to the public. By identifying landslide types using 3D topology (based on their shapes), our work highlights how important TDA can be in extracting meaningful insights about the failure types with just their shapes.

We want to bring the Reviewer’s attention that our work shows the application of TDA (a newly developed topology-based method), which is a topic of interest for mathematicians, data scientists, and researchers working on shape properties and classification. Moreover, our work has the potential for a broader impact on the geoscience community, as the method could be useful in varying geophysical domains (e.g., permafrost-borne retrogressive thaw slumps, submarine landslides, etc.) where unique landforms exhibit distinct shapes and configurations. By leveraging topology, we can better understand the mechanisms that govern these natural occurrences. Therefore, we believe Nature Communications is the best fit for the outreach of our work to a broader audience.

Data and methodology: The approach appears valid, although admittedly, I am no expert on topology as applied to 2 or 3-D datasets. The two landslide inventories used, the one from Italy and the other from the Pacific Northwest (Oregon and Washington), are certainly two of the larger datasets available and likely of high overall quality. I know it may not be standard in Nature articles to put the Methods section before the results, but I had a hard time interpreting the results until I skipped ahead to the Methods section. Consider placing the methods before the results and discussion. It makes it so much easier to understand and interpret the results.

Reply: We thank the reviewer for considering our approach valid. Regarding the concerns about section arrangement, we wish to clarify that the current structure adheres to the manuscript guidelines set forth by Nature Communications. Unfortunately, these guidelines constrain our ability to reposition the sections. We trust that this limitation is understandable, and we appreciate your consideration in this matter.

Analytical Approach: The analytical approach used in the investigation is robust and comprehensive. Suggested Improvements:

Line 19: Reference 1 - This article concerns landslide impacts in Germany. Why is it suitable for a statement about landslide damages worldwide?

Reply: We thank the reviewer for pointing this out. Indeed the article here refers to Germany but they mentioned worldwide landslide losses in Table 1 of their article. However, we understand that their claim came from their prior article, particularly, Klose et al. (2015) and we amend this reference to that of the latter. We thank the reviewer for this remark and we have amended it. Please find the changes in Line 19 in the revised manuscript.

Line 22-23: Reference to global annual landslide volume of “fifty-six billion cubic kilometers.” Is this the correct number? This would be the equivalent vol-

ume of filling up the Grand Canyon in the U.S. 13,440,000 times with landslide debris yearly (the Grand Canyon has a volume of 4,166 km³. Put another way; this reported volume of material would fill up the Caspian Sea (716,112,531 times EVERY YEAR)! This can not be correct. Do the authors mean fifty-six billion cubic meters, perhaps?

Reply: We thank the reviewer for this comment. Indeed, this was a mistake from our side. We should have used the units of cubic meters instead of cubic kilometers. Please find the amends in Line 23 in the revised manuscript.

Line 31: What is meant by the phrase “baffling predictive models?” Do you mean that the predictive models are baffling? As in, it is “baffling” that anyone uses the predictive model, or do you mean that the lack of information about failure mechanisms of mapped landslides baffles predictive models? Models are only as good as the data fed into them, so the phrase “baffling predictive models” confuses me as a native English-speaking reader.

Reply: Thank you for this comment. We are sorry that this term caused confusion in what we were trying to convey. By ‘baffling’, we simply meant that using unclassified landslide data to train predictive models can lead to biased predictions, as failures of different movement modes are combined singularly in one database. Therefore, ‘baffling’ here meant like getting perplexing (or confusing) results. We do realize that this wording might not have been too clear so we have amended this and you can find the amended text in Lines 31-34.

Lines 42-43. I do not follow the logic that suggests that because predictive models may indicate a low probability of landslide occurrence in a high landslide-prone region; therefore, the information needed to improve the model is identifying landslide failure mechanisms. The models could fail for many reasons. Garbage in = garbage out when it comes to models. I’m not convinced that the model fails because of a lack of landslide failure mechanism data rather than, for example, poor topographic or geologic data at the regional to global scale.

Reply: We thank the reviewer for highlighting the issue in this statement. The performance of landslide predictive models depends on topographic, geological, climatic, failure type, and other crucial information. Having incorrect information about one or more input parameters of predictive models will create bias in the models. So we expect that incorrect failure type will lead to bias in predictive models and that might lead to incorrect predictions. We used this line of logic when writing this statement. However, since the effect of a single input parameter on a predictive model is not clear or evident in past studies, this statement creates ambiguity and confusion. We thank the reviewer for pointing this out and we have decided to remove this particular statement from the revised manuscript to avoid this confusion.

Line 50-51: “Debris deposits at the talus” does not make sense. Talus is a deposit of rock debris at the base of a slope. The AGI Glossary definition of “talus” is Rock fragments of any size or shape (usually coarse and angular) derived from and lying at the base of a cliff or very steep, rocky slope.

Reply: We appreciate your attention to detail in noting the terminological

discrepancy related to "debris deposits at the talus." You are correct; according to the AGI Glossary definition, "talus" itself is a deposit of rock debris at the base of a slope or cliff. The phrasing was unintentionally redundant and potentially misleading. To rectify this, we have revised the text to eliminate the redundancy and to adhere more closely to standard geological terminology. We now simply refer to the "talus" when discussing the accumulation of rock fragments at the base of a slope or cliff, thereby aligning with the AGI Glossary definition. We have amended this in the entire manuscript, including the Figures and in the Supplementary Information. Thank you for bringing this to our attention, and we hope this revision provides greater clarity and terminological accuracy.

Line 87-88: "...deposited at the talus" does not make sense, especially for flow-type deposits, which usually are deposited on a debris fan (do you mean talus cone or fan?)

Reply: Similar to the previous comment, we have now rectified it according to the AGI glossary definition and amended the text to 'debris fan'. Please find the amended sentence in Line 93 in the revised manuscript..

Line 104: Fig. S1 -flows do not "follow the contours of the landscape. Rather, they cross contours, where contours represent lines of equal land-surface elevation. A more correct phrase would be "flows are the most sinuous, following the channelized topography more closely than other mass-wasting mechanisms."

Reply: Thank you for your insightful observation concerning the behavior of flows in relation to landscape contours. We acknowledge the inaccuracy in our initial phrasing that suggested flows "follow the contours of the landscape," which, as you correctly pointed out, is not consistent with the dynamics of mass-wasting processes. We appreciate your suggestion for a more precise phrase to describe this phenomenon. In the revised manuscript, we have opted for the phrase "flows exhibit the greatest sinuosity, following the channelized topography more closely than other mass-wasting mechanisms" We are thankful to the reviewer for enhancing the terminological and conceptual rigor of our work. Please find the amended text in Lines 109-111.

Line 204: What do you mean by "manually annotate a few instances to train the model?" By "few," do you mean 500? Still, quite a few to manually annotate.

Reply: Thank you for this comment. What we meant here was to digitize/label landslide polygon failure types manually, for example, using ArcGIS and Google Earth archive imagery. In the revised manuscript, we showed that even with 100 samples, the method achieves more than an average of 70 percent performance. For areas entirely lacking documented failure information, the model can be used by manually annotating a modest set of around 100 samples. This provides a practical solution for a streamlined, automated failure-type classification system in documenting failure types in landslide databases, often comprising thousands of samples (Please see transferability result sub-section).

Figures: Figure 1: For Type 1, "Talus" is misleading, as it does not need to be rock, which is what talus is composed of. "Debris" would be a better term. Also, for Type 1, "Scarp" is a landform, whereas "deposition" is a process. Thus

you are mixing a thing (landform) with a process. How about “erosion” and “deposition,” and then if you want to label the scarp, move it to the right side of the topographic profile (like you have for the label “talus.”)

Reply: We thank the reviewer for pointing this out. We have since changed Figure 1 according to the suggestions provided by the reviewer. Please find the changes in Figure 1.

Similar comment for the Type 2 Cross Section. How about changing “Scarp” to “Evacuation?” You already have “Scarp (at regolith)” on the left side of the topographic profile. No need to have it twice on the same diagram. In the caption, consider replacing the highlighted instance of “flows” with “moves” or “travels” since you already use the word “flow” two other times in the same sentence.

Reply: We thank the reviewer for this important comment. We have replaced the term ‘flows’ with that of ‘travels’ to avoid confusion between the action of ‘flow’ and the type of movement, ‘flow’ in the figure 1 caption.

What are the little blue symbols on the 3-D diagram next to Type 2 and 4 landslides? Are these rain clouds, lakes, or something else? Why show the houses and trees? Do the landslides impact them? Are they important to the different failure mechanisms highlighted in Figure 1? What does the dashed line just below the top of the “Steep slopes” polygon indicate? What is this important? Is the dashed line the steep slope break, or is the red polygon the area with steep slopes? Do you need both on the figure? Your Type 2 flow-type failure starts where slopes are not steep on the 3-D diagram.

Reply: We appreciate the reviewer’s insightful questions and will address each point in the following response.

The blue symbols representing rain clouds serve as symbolic triggers to illustrate potential initiating factors for flow-type or complex landslide behaviors. A reminder that the symbols are not meant to imply that rainfall is the sole trigger for such failures. Similarly, the depictions of houses and trees are included to symbolically underscore the potential impact of landslides on at-risk elements, such as infrastructure and natural vegetation. The aim is to emphasize the associated risks.

We acknowledge the reviewer’s concern about the use of dashed lines to indicate steep slope breaks. To avoid confusion, we have decided to eliminate the dashed lines and retain only the red polygons to denote steep slopes. You are correct in noting that the Type 2 failure does not initiate at the steep slope area. We apologize for the oversight and have corrected this issue in the updated Figure 1.

The revised Figure 1, which incorporates these changes, can be found in the updated manuscript. Thank you for helping us improve the clarity and accuracy of our work.

Figure 3: Part A’s caption reads, “The diagram also shows the 3D landslide polygon, which outlines the landslide shape.” Isn’t this a 2D polygon, the one on the right with the white center and black outline? I don’t see any 3D information associated with this part of the figure.

We appreciate the reviewer’s comment on this matter. The perception of the

polygon as 2D may arise from the angle at which the image was captured and the white background, which lacks reference points to highlight the polygon's 3D nature. We want to clarify that we utilized a 3D plotting library to generate this visual. We hope this addresses your concerns and thank you for allowing us the opportunity to clarify.

References: Yes. The manuscript references previous literature appropriately.

Line 398: What is the publication year for reference 21?

Reply: We apologize for the oversight in our bibliography. The year "2018" has now been added to the relevant reference. You can find the updated information under reference number 22 in the revised manuscript. Thank you for bringing this to our attention.

Supplemental Information (SI): S2. First Paragraph. "Accumulated debris at the talus" does not make sense. Talus is accumulated debris (mostly rock) at the base of a steep slope.

Reply: We apologize for the oversight once again. The relevant sentence has been amended in the Supplementary Information. For your convenience, we direct your attention to Supplemental Information (SI)-S2 where this correction has been implemented.

Fig. S1 - Caption: Flow-type failures [all failures] do not "follow the landscape's contour" they cross contours, where contours represent lines of equal land-surface elevation.

Reply: We thank the reviewer's comment on this matter. We have amended this in the caption of the figure and have chosen the phrase 'follow the landscape's channelized topography'

REVIEWER COMMENTS

Reviewer #1 (Remarks to the Author):

This research examines the identification of different types of landslides by using both geometric and topological parameters based on their polygon and underlying topographic data. The authors present a robust workflow for topological analysis and demonstrate the reliable accuracy of the automatic classification of landslide types. The updated version of the manuscript has shifted its focus to the various types of landslides, rather than the mechanisms of failure. Additionally, the authors have included case studies from China, Turkey, and Denmark using reported data. The revised manuscript appears to be well-constructed, and the concerns raised by the reviewer in the previous version have been mostly addressed.

Some small issues:

L69 The authors suggest that some landslides were verified using Google Earth images, but the details of this verification process are unclear. Could the authors provide more information on this process in the supplementary material?

Reviewer #2 (Remarks to the Author):

I appreciate the efforts by authors to improve the m/s. However, I am still not quite convinced about the transferability of the method. Thus I think the m/s isn't suitable to 'Nature communications'.

Reviewer #4 (Remarks to the Author):

Landslides are not only natural processes shaping the landscapes, but they are also hillslope agents that lead to a broad range of direct and indirect problems to the population, infrastructure and the environment (damaged roads and houses, soil degradation, displaced population, fatalities to name but a few). It is therefore not surprising that the prediction of where (susceptibility) new landslides may occur has been a research topic in itself for the past decades evolving a rather large scientific community.

Data-driven models are commonly used to assess the susceptibility of landslides. Although many models have been proposed with more or less advanced quantitative iterations framed, for example, around machine learning algorithms, one key gap in the whole prediction systems is related to the basic information about landslides that feed these models. Data-driven models rely indeed on landslide inventories. One common method drawback of these models as clearly highlighted by Rana and co-

authors is associated with the fact that the inventories commonly do not make any distinction between landslide types. In other words, the models are calibrated on a combination of processes that can be very different; which alters their predicting capability and potentially leads to meaningless outputs.

The absence of typology on slope failure processes in the inventories is clearly the result of the time investment needed to actually differentiate the types of landslides. In addition, such process typology identification requires a lot of expertise.

The study by Rana and co-authors proposes a new quantitative topology-based method built on the 3D morphology of the landslide shapes to infer about the typology of the slope failure processes. Developing a transferable approach based on limited data (landslide contours and digital topographic data), this research allows through landslide topology analysis to differentiate four groups of slope failure types, but also to get insight on the slope failure processes.

The findings of this well documented research are of wide interest. The implications with respect to landslide inventory, prediction and process characterisation, aspects that are well discussed in this work, are important. In addition, the significance of this work goes beyond this as it clearly points to the importance of looking at landslide/slope failure processes in general. It also applies a method (topology) that shows potentials for researchers working in classification topics for example.

The proposed data and methods and specifically the statistics and models are well justified and described. The supplementary material provides the information needed to make sure that the statistics are used in an appropriate way and uncertainties thereof well treated.

Overall the work of Rana and co-authors is timely, conceptually well designed, technically robust and of interest to a wide readership.

I have also appreciated the way they the authors have responded to the concerns addressed by the reviewers, especially with the fact that they have added in their analysis several other landslide inventories.

With respect to all the material provided in the manuscript and the response to the reviewers' comments; there are four rather minor comments/issues I would like to insist on so that it could help to better frame some statements, especially in the discussion/perspective parts:

- Landslide processes: four types of slope failures are identified, namely slides, flows, rock fall and complex. One very common landslide type that is observed in many regions are avalanches, and they are surely present in the inventories used in this study. Avalanches are a flow type process (Hung et al., 2014) but may occur in topographic conditions that are very similar to those of slides. What is the opinion of the authors about this issue? Could this be a caveat of their approach that must be mentioned?
- Landslide processes: the distinction between deep-seated and shallow landslides (Sidle and Bogaard, 2016) is a key property to take into account when analysing landslide failure types since it has a strong

implication with respect to the causes (for example vegetation characteristics that act mostly on shallow slope failures) and the triggers (response to rainfall and earthquake) at the origin of slope failure. The depth importance is particularly important in the context of the land transformation and the Anthropocene and it is a drawback of the landslide type classification schemes that do not really consider this aspect yet (Maki Mateso et al., 2023). Maybe a small discussion aspect related to the fact that the approach does not deal with that process differentiation could be added.

- The approach proposed by Rana and co-authors offers many perspectives. One of them is definitely about the characterization of landslide types and processes at large scale levels such as mentioned in line 265. However, manual mapping efforts at these scales are not common and one may expect that many inventories in the future are generated from automatic detection tools (Casagli et al., 2023). Although promising, such detected inventories provide shape delineations that are less accurate than manual ones. This is maybe something to add in the discussion.
- Among the perspectives options, how about applying the approach on landslide in Mars?

References

Casagli, N., Intrieri, E., Tofani, V., Gigli, G., Raspini, F., 2023. Landslide detection, monitoring and prediction with remote-sensing techniques. *Nature Reviews Earth and Environment* 4. doi:10.1038/s43017-022-00373-x

Hungr, O., Leroueil, S., Picarelli, L., 2014. The Varnes classification of landslide types, an update. *Landslides* 11, 167–194. doi:10.1007/s10346-013-0436-y

Maki Mateso, J., Biolders, C.L., Monsieurs, E., Depicker, A., Smets, B., Tambala, T., Bagalwa Mateso, L., Dewitte, O., 2023. Natural and human-induced landslides in a tropical mountainous region: the Rift flanks west of Lake Kivu (DR Congo). *Natural Hazards and Earth System Sciences* 23, 643–666. doi:https://doi.org/10.5194/nhess-23-643-2023

Sidle, R.C., Bogaard, T.A., 2016. Dynamic earth system and ecological controls of rainfall-initiated landslides. *Earth-Science Reviews* 159, 275–291. doi:10.1016/j.earscirev.2016.05.013

Author's response to comments

February 12, 2024

We thank the reviewers for their continued dedication to helping us improve our manuscript. We have made the necessary amends as requested, which we will follow up on in this response letter. We hope that the reviewers will find our revised manuscript suitable for publication. Normal font text is our responses to the reviewer comments (color-coded separately for each reviewer), whereas *italic text depicts changes or additions to the manuscript and the supplementary.*

Response to reviewer 1

This research examines the identification of different types of landslides by using both geometric and topological parameters based on their polygon and underlying topographic data. The authors present a robust workflow for topological analysis and demonstrate the reliable accuracy of the automatic classification of landslide types. The updated version of the manuscript has shifted its focus to the various types of landslides, rather than the mechanisms of failure. Additionally, the authors have included case studies from China, Turkey, and Denmark using reported data. The revised manuscript appears to be well-constructed, and the concerns raised by the reviewer in the previous version have been mostly addressed.

Reply: We thank the reviewer for their comments and appreciate that our revised version was received well.

Some small issues: L69 The authors suggest that some landslides were verified using Google Earth images, but the details of this verification process are unclear. Could the authors provide more information on this process in the supplementary material?

Reply: We agree with the reviewer's comment. As per the suggestion, we have added another section in the supplementary document highlighting how we verified the past 2005 and 2007 landslides in Wenchuan, China, using archived Google Earth imagery. We reproduce the material below, which can also be

Figure 1: *Panels a and b display model predictions for the 2005 inventory, while panels c and d pertain to the 2007 inventory. The analysis of certain features, like conspicuous or prominent scarps, channels, mountain ridges, and travel distances, aids in understanding failure movements. The plots illustrate predictions for both debris slides (2005: 14; 2007: 32) and debris flows (2005: 117; 2007: 40), using these features to corroborate and verify the two types of movements.*

found in Supplementary Section S5.

”To verify landslide movement types from Google Earth images, we focus on specific geomorphological features that are characteristic of different types of landslides (debris slides and debris flows in the case of Wenchuan, China). The approach is as follows:

- ***Examination of debris flow characteristics:***
To identify debris flows, we searched for channelized flow patterns [6]. This phenomenon is crucial in differentiating debris flows from other landslide types. Our attention was also directed towards the recognition of channels along the flow paths (Figure 1-a and c), indicative of channelized debris transportation, a hallmark of debris flow activity [9]. Furthermore, we also assessed the sinuosity and travel distances of these flows, as we suspect the hillslope channelized debris flow to harbor longer travel dis-

Figure 2: *Panels a and c show model predictions for landslides in 2005 and 2007, while panels b and d show landslides identified by Fan et al. (2019) after the 2008 Wenchuan earthquake. Archived Google Earth images are used to track landscape changes on slopes affected by landslides post 2005 and 2007.*

tances than debris slides [12].

- ***Examination of debris slide characteristics:***

We looked for abrupt, steep cliff-like features in the terrain where prominent scarps (see Figure 1-b and d) are often indicate debris slides and/or shallow slides [13]). Additionally, many debris slides were observed on the ridge of the mountains, necessitating further scrutiny of these landslides in such topographic conditions. Another key aspect of our analysis involved assessing the sinuosity and travel distance. Typically, debris slides exhibit less sinuous paths with shorter travel distances than debris flows [12]. Therefore, we looked for settings where such topographic conditions were met to qualify and confirm the movements as debris slides.

- ***Archive imagery to track landscape changes:***

We also employed the timeline feature of Google Earth images to conduct a more nuanced and comprehensive understanding of the individual landslides, their evolution, and what is occurring nearby in the topography to better understand the typology. We looked at the initiations, pathways, ac-

cumulations, terrain changes, and material deposition along the channels and hillslopes to better assess the movement type. By systematically examining historical imagery, we were able to verify, for example, the movement and accumulation of debris, providing more insights into verifying the predictions made by the model more effectively (see Figure 2).

Reviewer 2

I appreciate the efforts by authors to improve the m/s. However, I am still not quite convinced about the transferability of the method. Thus I think the m/s isn't suitable to 'Nature communications'.

Reply: We appreciate the reviewer's concern about the transferability of our method. Upon revisiting the concerns stemming from previous review comments, we realize that there is confusion regarding the term 'transferability'. We initially used the term 'method transferability' to imply the broader applicability of the overall method (topological data analysis with machine learning model) in diverse regions. This term reflected training and testing in the same region across the various inventories that we had worked with, and we wanted to showcase that the method, overall, worked well across the regions. However, we realize that this term can create an ambiguity where it can be confused to reflect 'model transferability', which implies training in one region and directly applying it to test the model in another region. Therefore, to remove this ambiguity and clarify the outcomes of our work, we amend the title and the text under the previous sub-section 'Method transferability to different regions of the world'. We appreciate the reviewer's valuable feedback in helping us to clarify the text on this matter.

We have now edited the text to three directions to outline our current results. These three directions or aspects are highlighted as 1) method's implementation in geomorphologically diverse regions with training and testing individually in each region, 2) method's implementation in geomorphologically diverse regions with limited or no failure-type information, and 3) method implementation in a region across multi-temporal inventories (temporal transferability). We also edited the title to reflect our results clearly; please see "Method implementation for failure-type identification" in the Results section.

- **Training and testing in regions individually:**

In the first aspect, we test the performance of our method's implementation across diverse geomorphological and climatic settings in the regions of the US Pacific Northwest (states of Oregon and Washington), Turkey, Denmark, and Wenchuan (China: 2008 earthquake event). This method evaluation scheme involved training and testing the method individually for each region, where the mean classification accuracy was observed above 80% for each region (see Figure 3-c). This result highlights the **adaptabil-**

Figure 3: Plot (a) shows the classification accuracy (in testing sample's index, and the y-axis shows the class probability corresponding to each failure class. Plot (b) shows the classification accuracy (in %) corresponding to each failure type with the number of training samples. The x-axis shows the number of training samples from each class used to train the model, and the y-axis shows the classification accuracy (in %) corresponding to each class. At 100 samples, the mean classification accuracy already reaches over 70% in Italy. Plot (c) demonstrates the model's versatility in accurately identifying various types of landslides across multiple geographical regions, including China, Turkey, Denmark, and the US Pacific Northwest, with their corresponding F1-scores listed for each type. Plot (d) highlights the model's capability to distinguish sub-types of landslide failures, specifically in the US Pacific Northwest and Italy. In the US Pacific Northwest, the model successfully classifies four additional sub-types—rotational slide, translational slide, debris flow, and earth flow—with an average F1-score of 84% along with the other failure type classes (complex and fall type). In Italy, the model identifies two additional sub-types—debris flows and earth flows—with an average F1 score of 96% along with the other failure type classes (slide, complex, and fall type).

ity of the proposed method in acclimating to these varying geographical regions.

- **Implementation in data-scarce contexts:**

In real-world scenarios, most landslide inventories have limited or no information regarding the failure-type. To evaluate the method performance in the regions with limited samples, we train and test the method with different amount of training samples. In the Italian case where the landslide samples are distributed around the country, even a small subset of samples <1% of the entire inventory (~100 samples from each class, see Figure 3-b) achieves more than >70% performance. Moreover, in a testing region like Wenchuan (see Supplementary Figure S6-a for Wenchuan) where the landslides are distributed in a basin, even with <20 samples from each class, it achieves more than 75% performance. This analysis highlights that irrespective of the scale (national or basin level), the model can perform even with limited training data. Similar results were also achieved in the US Pacific Northwest (see Supplementary Figure S6-b for the US) as well (>65% performance at ~100 samples from each class), demonstrating the method’s consistent effectiveness.

Similarly, in real-world scenarios where landslide databases are undocumented i.e., devoid of any information regarding failure types, landslide experts can train the model on a manually annotated small subset of samples and then classify the remaining landslides in the inventory. The amount of samples that landslides practitioners have to annotate will depend on the scale of the region; for example we show that in landslide databases in Italy that span the entire country, we can annotate less than <1% (~100 samples from each class) samples of the total data, and for Wenchuan region (at a basin-scale), we need <0.5% (~20 samples from each class) samples of the total, respectively.

Please note that training and testing in two disparate geographical regions can create bias because distinct failure types from different regions can have dissimilar shapes and thereby, dissimilar topological responses for the same failure movements. This challenge is due to differences in landscape features (both internally by geology and topography and externally by climate and tectonic activity) worldwide, and therefore, impacts how landslide propagate differently in different parts of the world. To train and test in different regions, we need a model that includes landslide instances spanning the diverse geomorphological settings across the globe. However, such a comprehensively curated dataset is currently unavailable (to the best of the author’s knowledge). Curating such comprehensive data is also difficult and time-expensive and moves beyond the scope of our work.

- **Implementation across multi-temporal inventories (temporal transferability):**

We implemented the method in an event-specific multi-temporal inventory from China, containing eight event-specific inventories from distinct years,

triggered by a combination of rainfall and earthquakes. To assess the method's performance for temporal transferability, we trained the model using the initial earthquake-triggered inventory from 2008 and then tested it on the remaining inventories from the subsequent years (2011 to 2018). Across all the testing inventories, our method achieved a mean classification accuracy of over 90% (see Figure 6-c, main manuscript), highlighting the temporal transferability of the model. Additionally, we implemented the method on two undocumented inventories predating the 2008 event, specifically from 2005 and 2007 (see Figure 6-a, b, main manuscript). In our prediction of the landslide types within these inventories, we found that most were identified as debris flows (117 in 2005 and 40 in 2007), with the remaining few classified as debris slides (14 in 2005 and 32 in 2007). We manually verified the method's predictions for these landslides using Google Earth archive imagery. Further details regarding the verification are in Supplementary Section S5 and Figures S3 and S4.

We refer the reviewer to the amended text in Lines 193-208. We have added the limitations of our approach in the Discussion here below (Lines 326-349). We also discussed the perspectives towards predicting new emerging movement types that the model never experienced. We sincerely hope that the reviewer finds our comprehensive response and the subsequent modifications (highlighting the limitations) to the manuscript convincing and reflective of acceptance at Nature Communications.

"While the proposed method demonstrates notable success in identifying failure types, there are inherent limitations. Although the method adapts across regions (US Pacific Northwest, Turkey, Denmark, and Wenchuan in China) and temporal settings (Wenchuan, China), the accuracy of the results is limited by the specific landslide movement types available during training. Hence, all new landslide types must be introduced to the model in advance. We showcased that the model can easily accommodate additional landslide types if desired (see results of sub-type movements in Results: "Determining sub-type failure movements of the landslides", Figure 5-d). We suggest continual updates to both the model and inventories with new movement types as they become available to sustain reliable performance.

Another concern about our approach is transferability. An ideal solution would be a transferable model that can be trained and tested in geographically disparate areas. However, the development of such an ideal solution is exceptionally challenging. Model training and testing in such disparate conditions can introduce bias since distinct failure types from different regions can have dissimilar shapes and thereby, dissimilar topological responses, even for the same failure movements[9]. This challenge is due to diversities in the landscape features, both internally by geology and topography and externally by climate and tectonic activity worldwide and therefore, impacts how landslides propagate in varying ways in different parts of the world. Alternatively, comprehensive training landslide data that encompasses the diverse landscapes of the earth can aid in achieving a

global generalization. However, such detailed-curated data is unavailable at the moment.

In real-world scenarios where the landslide databases are undocumented, i.e., devoid of any information regarding failure types, practitioners and experts can train the model on a manually annotated small subset of samples and then classify the remaining landslides in the inventory using our method. The amount of samples that need to be annotated depends on the scale of the region. For example, in Italy, ~100 samples from each class achieved satisfactory results, whereas in Wenchuan, China, <20 samples from each class were enough. This practical solution ensures an economical direction to obtain reasonable prediction capability in new study areas (see Figure 5-b and Supplementary Figure S6). Additionally, we chose the combination of TDA with decision tree-based shallow learning[8] (i.e., random forest) deliberately to capitalize on a functioning model for data-scarce contexts to ensure practicality in the real world. This solution expresses a streamlined approach highlighting the importance of dynamic, adaptable models in improving landslide prediction across diverse regions.”

Reviewer 4

Landslides are not only natural processes shaping the landscapes, but they are also hillslope agents that lead to a broad range of direct and indirect problems to the population, infrastructure and the environment (damaged roads and houses, soil degradation, displaced population, fatalities to name but a few). It is therefore not surprising that the prediction of where (susceptibility) new landslides may occur has been a research topic in itself for the past decades evolving a rather large scientific community.

Data-driven models are commonly used to assess the susceptibility of landslides. Although many models have been proposed with more or less advanced quantitative iterations framed, for example, around machine learning algorithms, one key gap in the whole prediction systems is related to the basic information about landslides that feed these models. Data-driven models rely indeed on landslide inventories. One common method drawback of these models as clearly highlighted by Rana and co-authors is associated with the fact that the inventories commonly do not make any distinction between landslide types. In other words, the models are calibrated on a combination of processes that can be very different; which alters their predicting capability and potentially leads to meaningless outputs.

The absence of typology on slope failure processes in the inventories is clearly the result of the time investment needed to actually differentiate the types of landslides. In addition, such process typology identification requires a lot of expertise.

The study by Rana and co-authors proposes a new quantitative topology-based method built on the 3D morphology of the landslide shapes to infer about the typology of the slope failure processes. Developing a transferable approach based on limited data (landslide contours and digital topographic data), this research allows through landslide topology analysis to differentiate four groups of slope failure types, but also to get insight on the slope failure processes.

The findings of this well documented research are of wide interest. The implications with respect to landslide inventory, prediction and process characterisation, aspects that are well discussed in this work, are important. In addition, the significance of this work goes beyond this as it clearly points to the importance of looking at landslide/slope failure processes in general. It also applies a method (topology) that shows potentials for researchers working in classification topics for example.

The proposed data and methods and specifically the statistics and models are well justified and described. The supplementary material provides the information needed to make sure that the statistics are used in an appropriate way and uncertainties thereof well treated.

Overall the work of Rana and co-authors is timely, conceptually well designed, technically robust and of interest to a wide readership.

I have also appreciated the way they the authors have responded to the concerns addressed by the reviewers, especially with the fact that they have added in their analysis several other landslide inventories. With respect to all the material provided in the manuscript and the response to the reviewers' comments; there are four rather minor comments/issues I would like to insist on so that it could help to better frame some statements, especially in the discussion/perspective parts.

Reply: We thank the reviewer for appreciating our work and revisions for the previous round of reviews. We acknowledge the reviewer for the perspectives that they provided to the Discussion section. We believe that these perspectives are essential to not only highlight the importance of our work but also to emphasize the next steps that can be taken in the community to advance the work towards process understanding and expanding to other geophysical and extraterrestrial domains such as Martian landslides. We show the addressed remarks point-by-point in the next steps.

1. Landslide processes: four types of slope failures are identified, namely slides, flows, rock fall and complex. One very common landslide type that is observed in many regions are avalanches, and they are surely present in the inventories used in this study. Avalanches are a flow type process (Hung et al., 2014) but may occur in topographic conditions that are very similar to those

of slides. What is the opinion of the authors about this issue? Could this be a caveat of their approach that must be mentioned?

Reply: We thank the reviewer for raising an important question concerning the classification of failure types that overlap the categories in our study (slides, flows, falls, and complex). Debris avalanches, while classified as flow-type processes, often manifest under topographical conditions akin to those of debris slides, as noted by the reviewer. We have provided our insights based on topological analysis and its ability to distinguish debris avalanches from debris slides, while also commenting on conditions when separating the two types might be difficult. We believe that this question regarding avalanches raises an important discussion that warrants mention. However, due to limited space in our Discussion section, we briefly addressed this topic in Lines 293-297 in the discussion and provided a detailed exploration in Supplementary Section S6 entitled, 'Topology and outlook towards debris avalanches', inviting readers to consult this extended analysis.

Discussion Section: Lines 293-297

Drawing on the experiments of the sub-type classification of landslide failures in Italy and the US Pacific Northwest, we recognize the substantial value in identifying and quantifying these sub-movements, bearing notable potential to enhance both landslide risk assessment and related hazard models²⁰. The level of damage to infrastructures and the risk of human casualties vary depending on the intensity of the failure movement, which differs for each failure type¹⁴. For example, a slow-moving deep-seated rotational landslide (1.5 m/year to 16 mm/year) may not pose an immediate threat to the population, but it can cause extensive structural damage to buildings over a prolonged period^{47,48}. In contrast, flow-type failures, such as debris flows, have rapid mobility and can result in significant casualties and infrastructure damage simultaneously^{49,50}. Similarly, episodic impacts in fall-type failures can cause massive damage to infrastructures in a matter of seconds due to their high energy (e.g., impact pressure measured in kilopascals, kPa)⁵¹. We can infer from these broad examples that the availability of failure-type, especially sub-type information could considerably enhance the accuracy of predictive modeling and that incorporating it benefits the landslide community as it enables the development of accurate landslide predictive models. *Nevertheless, certain failure movements fall in between, for instance, avalanches (flow-type processes) between slides and flows, where avalanches can occur in similar topographical conditions as slides. The presence of larger boulders in avalanches than debris flows has concrete implications for the protective measures required to absorb the impacts. We explore this facet, particularly, separating the intertwined movements of avalanches from debris slides, in Supplementary Section S6.*

Supplementary Section S6:

”Debris avalanches, categorized within the realm of flow-type movements, occur often in topographical settings similar to those conducive to debris slides (Hungre et al. 2014[9]). Consequently, the morphological and material characteristics of debris slides and debris avalanches may exhibit notable similarities. This observation is corroborated by the fact that debris slides are frequently categorized as a preliminary stage in the development of debris avalanches (and also debris flows), serving as a mechanism of initiation (Hungre et al. 2014[9]). In the context of debris avalanches, we anticipate that their movement will impart a topographic signature that, while generally akin to that of debris slides, remains distinct. Specifically, debris avalanches are characterized by the formation of a debris fan (conical shape) at the deposition part, a feature amiss in debris slides. These avalanche deposits have distinct imprints on the landscape, thereby influencing the topology of the landslide polygonal shape, which Topological Data Analysis (TDA) can effectively discern. Although the initial stages of both debris slides and avalanches may exhibit similar morphologies, this variation in the deposition area of the landslide can be leveraged by our model to distinguish between debris slides and debris avalanches. In consensus, as long as there are distinct variations in shapes amongst the different movement types, our model can distinguish them, even between debris slides and avalanches.

A caveat in the approach can stem when the performance of the model is conditioned on the preciseness of the avalanches mapped in the inventory. These inventories are primarily sourced/generated from expert observations (both in the field and remotely) and institutional reports. This reliance may inadvertently introduce a bias, particularly in differentiating between landslide types with subtle distinctions in their topographical and material characteristics. A pertinent example could be the challenge of separating debris avalanches from debris slides.

Largely, debris slides are grouped as an initial component of debris flows or debris avalanches, which they function as an initiation mechanism (Hungre et al., 2014[9]). This feature truly indicates that landslides are quite complex by nature where a failure event could manifest more than one mode of movement (for example, a landslide transitioning from initial soil cracking to a debris avalanche and finally into a surging sediment-laden channel, collectively termed as ‘debris flow’). Our model, therefore, tends to classify landslides based on their most dominant or final observable state, potentially simplifying the multifaceted dynamics of landslide processes. For instance, a debris slide evolving into a debris avalanche might predominantly be identified as the latter by our model since that would be the last imprint on the topography.

To address this limitation, a more nuanced model could be developed, focusing on initial indicators such as the scarp area, which may provide insights into the primary failure mechanism, thereby enhancing the model’s classification ability (Baron et al., 2024[1]). Alternatively, we can adapt existing data to consider the evolution stages (capturing and separating the moment when debris slides converge to an avalanche) but this would take more effort in re-curating the data to avoid misclassifications.”

2. Landslide processes: the distinction between deep-seated and shallow landslides (Sidle and Bogaard, 2016) is a key property to take into account when analysing landslide failure types since it has a strong implication with respect to the causes (for example vegetation characteristics that act mostly on shallow slope failures) and the triggers (response to rainfall and earthquake) at the origin of slope failure. The depth importance is particularly important in the context of the land transformation and the Anthropocene and it is a drawback of the landslide type classification schemes that do not really consider this aspect yet (Maki Mateso et al., 2023). Maybe a small discussion aspect related to the fact that the approach does not deal with that process differentiation could be added.

Reply: We thank the reviewer for this important comment. The distinction that our approach provides from the process of a landslide is crucial to define since our approach does not deal with the process or mechanism side of landslides. Instead, our focus is centered on understanding landslide failure movements through a three-dimensional topological lens. This perspective allows us to examine the spatial configuration and characteristics of landslides beyond their mere shapes. However, linking these shapes to understand processes is naturally the next step, which does warrant an important outlook. We have therefore added a new discussion regarding this, which can be found in Lines 310-325 in the Discussion section.

"The 3D topological descriptions of landslides that we use do not directly connect landslide failure movement to the failure process, as we rely solely on the topography and the shape of the landslide. In establishing characteristics for failure movements with topological information, the model provides insights into the predominant landslide types prevalent in the landscape. Regardless, it's important to recognize that 3D-based topology alone may not directly lead to the underlying processes driving landslides. Incorporating ancillary information such as land use characteristics (e.g., vegetation or lithology) and information on the triggers (e.g., peak-ground acceleration or rainfall intensity) into future models could tailor the characterization of mapped landslides to assess particular failure processes.

Our approach has the potential to link the movement types that emerge from anthropogenic activities with features that may deviate from naturally induced landslides. For instance, fill slope landslides induced by human activities might resemble naturally-induced deep-seated landslides in appearance, but their underlying processes are distinctly different. This overlap poses challenges to effectively comment on the processes behind landslide failures (between urban and natural landslides) that look similar.

Anthropogenic activities such as land use change, urbanization, and soil management practices have various impacts on landslide activities, sometimes leading to complex scenarios (Dille et al., 2022[7]). If trained with appropriate data, our model could effectively distinguish between landslide types in urban and natural

environments and be instrumental in studying the disparities in between. By doing so, our method can improve mapping and classification protocols, which may contribute to understanding how human pressures, particularly in urban areas, are influencing the timing and characteristics of landslide failures (Maki Mateso et al., 2023[10]).”

3. The approach proposed by Rana and co-authors offers many perspectives. One of them is definitely about the characterization of landslide types and processes at large scale levels such as mentioned in line 265. However, manual mapping efforts at these scales are not common and one may expect that many inventories in the future are generated from automatic detection tools (Casagli et al., 2023). Although promising, such detected inventories provide shape delineations that are less accurate than manual ones. This is maybe something to add in the discussion.

Reply: We appreciate the reviewer’s insightful observation. Indeed, it is crucial to acknowledge that the current advanced automated tools and models, such as deep learning algorithms (e.g., U-Net, vision transformers), offer rapid mapping of both historical, recent event-based, and multi-temporal landslides (Casagli et al., 2023[3]; Behling et al., 2016[2]). Nevertheless, the issue with landslide shape delineation still remains a big problem with such tools, especially the problem of amalgamation in landslides (Marc et al., 2015 [11]). Imprecise delineation of landslide bodies, in particular, the prediction of two or more landslides as one entity by these automated tools, is a big issue that can cause difficulty in identifying the right failure movement by our model. Before adapting our model to the outputs of such automated tools, the problem related to amalgamations must be addressed. We are thankful for the reviewer’s perspective in this regard. At the same time, due to the limited space in the Discussion section, we have added the elaborated text in the Supplementary Section S7, ‘Technical limitations of the method’ that discusses the technical limitations of the method.

”Certain limitations can arise pertaining to the shape delineation of landslide bodies. Manual mapping efforts provide the most accurate representation of landslides when mapped from field or remote observations when compared to automated mapping. In the past years, despite automated tools gaining traction due to their reliability in rapid mapping and assessment (Casagli et al., 2023[3]), their use in generating complete inventories (either event-based or historical) is still limited. These limitations arise from issues like amalgamations (Marc et al., 2015[11]) where more than one landslide body is mapped together as a single entity due to similar spectral responses of the landslide pixels in satellite images. Inadvertently, it affects the assessment of reliable landslide statistics such as travel distance, propagation area, and volume estimates. For our model to operate properly, the amalgamation issue needs to be addressed as an intermediate step following mapping by automated tools. An avenue in this endeavor would be to explore the sensitivity between manual and automated inventories

via TDA.”

4. Among the perspectives options, how about applying the approach on landslide in Mars?

Reply: Thank you for this comment. We believe this is an excellent domain to explore, particularly since a lot of momentum has been gained regarding Martian landslides in the past few years. With respect to topological data analysis, we anticipate that the varying failure movements on the Martian surface such as rock avalanches or slump-flows, would exhibit diverse geometrical shapes, which can be leveraged for failure-movement identification via topology. Integrating the topological information of these extraterrestrial landslides with its geomorphology could assist in understanding the paleoenvironmental conditions of the planetary surfaces since landslide mobility and their characteristics provide insights into rock properties and the conditions of sediment deposition at the time of their occurrence [4, 5].

We were very excited to look into this aspect directly, and hence, we searched for properly curated landslide data and DEM to test our model on Mars. Unfortunately, we were not able to find polygon-based landslide data on Mars to test out our model and involve this side of the conversation in the discussion more explicitly. Regretfully, we could only mention a short text in this regard superficially. This addition definitely enriches the discussion section and paves a possible outreach to the broader Martian geophysical community. We add the relevant text to the Discussion section in Lines 357-361.

”Furthermore, the lens of topological characteristics can enable research on extra-terrestrial landslides such as those on Mars. The topology of Martian rock avalanches, slumps, and slump-flows with cryosphere characteristics (Crosta et al., 2018[4]) can assist in understanding the paleoenvironmental conditions on the planet. Characteristics of landslide mobility could provide insights into material properties and the conditions of sediment deposition at their occurrence time, for example, the presence of water and ice content [4, 5]”

Minor comments:

1. Figure 1: here you mention ”soil”, while you use ”regolith” in the figure. To be harmonized?

Reply: Thank you for this comment. We agree that it is better to harmonize the text so as to not cause confusion for the reader. We have hence amended the text, which you can find in the caption of Figure 1.

2. Line 62: IFFI- this acronym is defined later in the édata availability” section. Nevertheless, it could be defined here.

Reply: We have made the necessary changes and we refer you to Line 63 for

the amends.

3. Figure 4: the full name is used in the figure of the supplementary material. here, in Figure 4, the abbreviations are used. Maybe that is to be harmonized.

Reply: We agree with the reviewer that a harmonization regarding the abbreviations is required to maintain consistency. Hence, we have changed the labels into the associated abbreviations in the supplementary material in Figure S2.

4. Figure 4 caption: this is not clear: 'from top to bottom'.

Reply: The phrase 'from top to bottom' is confusing and in retrospect, it's also not required to mention. Hence, we decided to remove this phrase.

5. Figure 4: refer to supplementary material for the definitions.

Reply: We added the sentence "*(please refer to the supplementary section S4 for their definitions)*" for your reference in Figure 4, which contains the definitions.

6. Line 144: BC_C

Reply: We thank the reviewer for pointing this mistake out. Indeed, the topological feature here is BC_C and not BA_C .

7. Figure 5: Most commonly, this is spelled in one word: earthflow (see Hungr et al., 2014 in Landslides for example)

Reply: We agree that the word "earthflow" is a common word, as also seen in Hungr et al. (2014)[9]. We have hence amended this to be a single word in the manuscript.

8. Line 227: May be I have missed something, but why not using the Italian inventory here?

Reply: We thank the reviewer for this comment. The reason we did not use the inventory from Italy here was that the Italian inventory did not contain information regarding the complex landslides as in the US counterpart. The latter contained ancillary notes regarding each complex landslide, such as "Translational rock slides followed by rock falls" or "Rotational slides followed by flows". However, due to the lack of such ancillary information, we could not employ this in Italy. Nevertheless, we look forward to using more complex landslides in our future work to further expand into decoupling their movements and identify the coupled failure types within them.

9. Lines 382-383 to be consistent, I would also provide the number of landslides contained in this inventory.

Reply: Thank you for this comment. We have added the number of landslides for Wenchuan, China, to remain consistent in our writing.

10. Typos in the references

Reply: We have corrected the typos in the references as pointed out by the reviewer.

References

- [1] I. Baroň, J. Jelének, J. Klimeš, J.-J. Dong, R. Melichar, M. Šutjak, Y. Chen, C.-M. Yang, E.-L. Zhang, J. Méndez, C.-H. Tseng, F. Hartvich, J. Blahût, T.-T. Nguyn, L. Kociánová, F. Bárta, V. Dušek, and P. Kycl. Source area morphometry and high depletion rate of landslides may indicate their coseismic origin. *Engineering Geology*, 330:107424, 2024.
- [2] R. Behling, S. Roessner, D. Golovko, and B. Kleinschmit. Derivation of long-term spatiotemporal landslide activity—a multi-sensor time series approach. *Remote Sensing of Environment*, 186:88–104, 2016.
- [3] N. Casagli, E. Intrieri, V. Tofani, G. Gigli, and F. Raspini. Landslide detection, monitoring and prediction with remote-sensing techniques. *Nature Reviews Earth & Environment*, 4(1):51–64, 2023.
- [4] G. Crosta, P. Frattini, E. Valbuzzi, and F. De Blasio. Introducing a new inventory of large martian landslides. *Earth and Space Science*, 5(4):89–119, 2018.
- [5] G. B. Crosta, F. V. De Blasio, and P. Frattini. Global scale analysis of martian landslide mobility and paleoenvironmental clues. *Journal of Geophysical Research: Planets*, 123(4):872–891, 2018.
- [6] V. D. J. T. A. Cruden, David M. and R. Schuster. Landslides: investigation and mitigation. *Transportation research board special report*, 247:36–75, 1996.
- [7] A. Dille, O. Dewitte, A. L. Handwerker, N. d’Oreye, D. Derauw, G. Ganza Bamulezi, G. Ilombe Mawe, C. Michellier, J. Moeyersons, E. Monsieurs, et al. Acceleration of a large deep-seated tropical landslide due to urbanization feedbacks. *Nature Geoscience*, pages 1–8, 2022.
- [8] L. Grinsztajn, E. Oyallon, and G. Varoquaux. Why do tree-based models still outperform deep learning on typical tabular data? *Advances in Neural Information Processing Systems*, 35:507–520, 2022.

- [9] O. Hungr, S. Leroueil, and L. Picarelli. The varnes classification of landslide types, an update. *Landslides*, 11(2):167–194, 2014.
- [10] J.-C. Maki Mateso, C. L. Biielders, E. Monsieurs, A. Depicker, B. Smets, T. Tambala, L. Bagalwa Mateso, and O. Dewitte. Characteristics and causes of natural and human-induced landslides in a tropical mountainous region: the rift flank west of lake kivu (democratic republic of the congo). *Natural Hazards and Earth System Sciences*, 23(2):643–666, 2023.
- [11] O. Marc and N. Hovius. Amalgamation in landslide maps: effects and automatic detection. *Natural Hazards and Earth System Science*, 15(4):723–733, 2015.
- [12] B. Strîmbu. Modeling the travel distances of debris flows and debris slides: quantifying hillside morphology. *Annals of Forest Research*, pages 119–134, 2011.
- [13] D. J. Varnes. Slope movement types and processes. *TRB Special Report*, 176:11–33, 1978.